# Vector Quantization using Gaussian Variational Autoencoder

## Abstract

Vector quantized variational autoencoder (VQ-VAE) is a discrete auto-encoder that compresses images into discrete tokens. It is difficult to train due to discretization. In this paper, we propose a simple yet effective technique, dubbed **Gaussian Quant (GQ)**, that converts a Gaussian VAE with certain constraint into a VQ-VAE without training. GQ generates random Gaussian noise as a codebook and finds the closest noise to the posterior mean. Theoretically, we prove that when the logarithm of the codebook size exceeds the bits-back coding rate of the Gaussian VAE, a small quantization error is guaranteed. Practically, we propose a heuristic to train Gaussian VAE for effective GQ, named target divergence constraint (TDC). Empirically, we show that GQ outperforms previous VQ-VAEs, such as VQGAN, FSQ, LFQ, and BSQ, on both UNet and ViT architectures. Furthermore, TDC also improves upon previous Gaussian VAE discretization methods, such as TokenBridge. The source code is provided in supplementary materials.

## 1 Introduction

Vector-quantized variational autoencoder (Van Den Oord et al., 2017) is an autoencoder that compresses images into discrete tokens. It is fundamental to autoregressive generative models (Esser et al., 2021; Chang et al., 2022; Yu et al., 2023; Sun et al., 2024b). However, VQ-VAE is difficult to train: the encoding process of VQ-VAE is not differentiable and challenges such as codebook collapse often emerge (Sønderby et al., 2017). Special techniques are required to ensure the convergence of VQ-VAE, such as commitment loss (Van Den Oord et al., 2017), expectation maximization (EM) (Roy et al., 2018), Gumbel-Softmax (Jang et al., 2016; Maddison et al., 2016; Sønderby et al., 2017), and entropy loss (Yu et al., 2023; Zhao et al., 2024).

In this paper, we circumvent the challenge of training VQ-VAE by converting a Gaussian VAE with certain constraint into a VQ-VAE without any training. More specifically, we propose **Gaussian Quant (GQ)**, a simple yet effective method for training-free conversion. The core idea is to generate a codebook of one-dimensional Gaussian noise and, for each dimension of the posterior, select the codebook entry that is closest to the posterior mean. Theoretically, we show that as the logarithm of the codebook size exceeds the bits-back coding bitrate (Hinton & Van Camp, 1993; Townsend et al., 2019) of the Gaussian VAE, the resulting quantization error is small. In other words, GQ and the Gaussian VAE exhibit similar rate-distortion performance. This result serves as the theoretical foundation of GQ and provides a principled guideline for selecting codebook sizes.

Practically, we introduce the target divergence constraint (TDC) to train a Gaussian VAE for efficient conversion. TDC encourages the Gaussian VAE to achieve the same Kullback–Leibler (KL) divergence for each dimension, corresponding to the bits-back coding bitrate. Empirically, we demonstrate that GQ with Gaussian VAE trained by TDC, outperforms previous VQ-VAEs such as VQGAN, FSQ, LFQ, and BSQ (Van Den Oord et al., 2017; Mentzer et al., 2023; Yu et al., 2023; Zhao et al., 2024) in terms of reconstruction quality, using both UNet and ViT backbones. Additionally, we show that TDC can improve previous Gaussian VAE discretization methods, such as TokenBridge (Wang et al., 2025).

Our contributions can be summarized as follows:

- (Section 3.1) We propose GQ, a simple yet effective approach that converts a pre-trained Gaussian VAE with certain constraint into VQ-VAE without training.

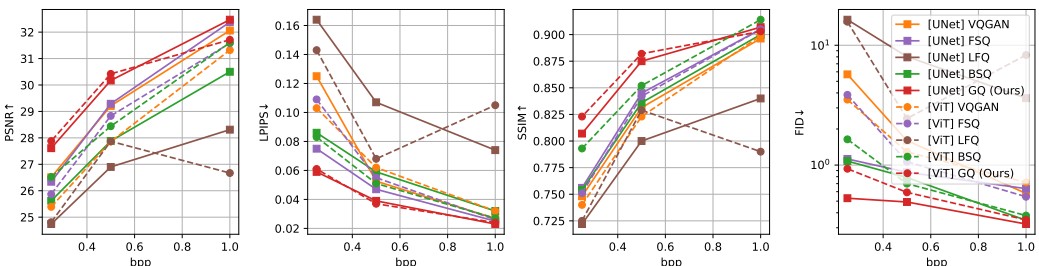

Figure 1: The rate-distortion performance on the ImageNet dataset demonstrates that GQ outperforms previous VQ-VAEs on both UNet and ViT architectures.

- (Section 3.2) Theoretically, we prove that when the codebook size of GQ is close to the bits-back coding bitrate of the Gaussian VAE, the conversion error remains small.
- (Section 3.3) Empirically, we introduce target divergence constraint (TDC) to implement GQ and show that GQ outperforms previous VQ-VAEs, such as VQGAN, FSQ, LFQ, and BSQ, on both UNet and ViT architectures.
- (Section 3.5) Furthermore, we show that TDC can be used to improve previous Gaussian VAE discretization approaches, such as TokenBridge.

## 2 PRELIMINARIES

We denote the original image as $X$, the latent variable as $Z$, the encoder as $f(\cdot)$, and the decoder as $g(\cdot)$. We use $\log(\cdot)$ to denote the natural logarithm (base $e$) and KL divergence as $D_{KL}(\cdot||\cdot)$. Similarly, we use $\log_2(\cdot)$ to denote the logarithm with base 2, and KL divergence as $D_{KL(2)}(\cdot||\cdot)$.

### 2.1 VECTOR QUANTIZED VARIATIONAL AUTOENCODER

VQ-VAE (Van Den Oord et al., 2017) transforms source image into a series of integer tokens, which can be decoded using a codebook and decoder. To facilitate auto-regressive generation, it typically involves a deterministic transformation and a shared codebook across different tokens. More specifically, VQ-VAE maintains a codebook $c_{1:K}$ with size $K$ and a bitrate of $\log K$. The encoding process of VQ-VAE involves finding the closest codeword $c_j$ in $c_{1:K}$ to the encoder output $f(x)_i$ for each latent dimension $i$. Denote distortion as $\Delta(\cdot, \cdot)$, the optimization target of VQ-VAE is the rate-distortion function weighted by Lagrangian multiplier $\lambda$:

$$\mathcal{L}_{VQ} = \lambda \underbrace{\log K}_{\text{bitrate}} + \underbrace{\mathbb{E}[\Delta(X, g(\hat{z}))]}_{\text{distortion}} + \mathcal{L}_{Reg},$$

$$\hat{z}_i = \arg\min_{c_j \in \{c_{1:K}\}} ||f(x)_i - c_j||, \text{ where } c_{1:K} \text{ are learned codebook,} \quad (1)$$

and $\mathcal{L}_{Reg}$ are regularization for VQ-VAE to converge, such as a combination of commitment loss and codebook loss (Van Den Oord et al., 2017) and Gumbel Softmax loss (Sønderby et al., 2017).

### 2.2 GAUSSIAN VARIATIONAL AUTOENCODER AND BITS-BACK CODING

The Gaussian VAE is a special type of VAE (Kingma et al., 2013) with a prior $\mathcal{N}(0, I)$ and a fully factorized Gaussian posterior $q(Z|X)$. The encoding process of a Gaussian VAE simply involves sampling the latent variable $z_i \sim q(Z_i|X)$ for each latent dimension $i$. Assuming $\log p(X|Z = z) \propto (1/\lambda)\Delta(X, g(z))$, then the negative evidence lower bound (ELBO) of the Gaussian VAE is equivalent to a rate-distortion function of a bits-back coding bitrate term and a distortion term:

$$\mathcal{L}_{VAE} = \lambda \underbrace{D_{KL}(q(Z|X)||\mathcal{N}(0, 1))}_{\text{bits-back coding bitrate}} + \underbrace{\mathbb{E}[\Delta(X, g(z))]}_{\text{distortion}},$$

$$z_i \sim q(Z_i|X = x) = \mathcal{N}(\mu_i, \sigma_i^2), i = 1 \dots d. \quad (2)$$

The bitrate of $z_i$ is the **bits-back coding bitrate**, defined as $D_{KL}(q(Z_i|X)||\mathcal{N}(0, I))$ (Hinton & Van Camp, 1993; Townsend et al., 2019). This is because, when compressing $X$ losslessly, one can communicate $z_i$ using $D_{KL}(q(Z|X)||\mathcal{N}(0, I))$ nats for arbitrary precision.

## 3 GAUSSIAN QUANT: VECTOR QUANTIZATION USING GAUSSIAN VAE

### 3.1 DIRECT QUANTIZATION OF GAUSSIAN VAE

We propose an extremely simple technique to obtain a VQ-VAE from a Gaussian VAE: we directly generate one-dimensional Gaussian noise as the codebook for VQ-VAE (Van Den Oord et al., 2017) and quantize the posterior mean $\mu_i$ of the Gaussian VAE independently for each dimension $i$. Because the codebook consists entirely of samples from a Gaussian distribution, we refer to our approach as **Gaussian Quant (GQ)**. Specifically, we randomly generate $K$ codebook values $c_{1:K} \sim \mathcal{N}(0, 1)$, which is the same for each dimension. Then, for each dimension $i$, we select the $c_j$ that is closest to the posterior mean $\mu_i$ and denote the quantized value as $\hat{z}_i$:

$$\hat{z}_i = \arg \min_{c_j \in \{c_{1:K}\}} ||\mu_i - c_j||, \text{ where } c_{1:K} \sim \mathcal{N}(0, 1). \tag{3}$$

### 3.2 THEORETICAL RELATIONSHIP BETWEEN THE CODEBOOK SIZE AND QUANTIZATION ERROR

Why GQ works and how to select $K$ are not straightforward questions. Theoretically, we show that GQ preserves the rate-distortion property of the Gaussian VAE: when the bitrate $\log K$ matches the bits-back coding bitrate of the Gaussian VAE, the quantization error is small. More specifically, we show that the probability of large quantization error decays doubly exponentially as the codebook bitrate $\log K$ exceeds the bits-back coding bitrate $D_{KL}(q(Z_i|X)||\mathcal{N}(0, 1))$.

**Theorem 1.** *Denote the mean and standard deviation of $q(Z_i|X = x)$ as $\mu_i$ and $\sigma_i$, respectively. Assuming that the product and sum satisfy $|\mu_i\sigma_i| \leq c_1$ and $|\mu_i| + |\sigma_i| \leq c_2$, the probability of a quantization error $|\hat{z}_i - \mu_i| \geq \sigma_i$ decays doubly exponentially with respect to the number of nats $t$ by which the codebook bitrate $\log K$ exceeds the bits-back coding bitrate. i.e.,*

$$\text{when } \log K = D_{KL}(q(Z_i|X)||\mathcal{N}(0, 1)) + t,$$

$$\Pr\{|\hat{z}_i - \mu_i| \geq \sigma_i\} \leq \exp(-e^t \underbrace{\sqrt{\frac{2}{\pi}} e^{-c_1 - 0.5}}_{\text{constant}}). \tag{4}$$

Conversely, when the codebook bitrate $\log K$ is smaller than the bits-back coding bitrate, the probability of large quantization error increases exponentially toward 1. More specifically, we show that the probability of large quantization error increases exponentially when the codebook bitrate $\log K$ is lower than the bits-back bitrate $D_{KL}(q(Z_i|X)||\mathcal{N}(0, 1))$.

**Theorem 2.** *The probability of a quantization error $|\hat{z}_i - \mu_i| \geq \sigma_i$ increases exponentially with respect to the number of nats $t$ by which the codebook bitrate $\log K$ is lower than the bits-back coding bitrate, i.e.,*

$$\text{when } \log K = D_{KL}(q(Z_i|X)||\mathcal{N}(0, 1)) - t,$$

$$\Pr\{|\hat{z}_i - \mu_i| \geq \sigma_i\} \geq 1 - e^{-t} \underbrace{\sqrt{\frac{2}{\pi}} e^{0.5c_2^2 - 0.5}}_{\text{constant}}. \tag{5}$$

Theorems 1 and 2 provide a principled guideline for choosing $K$, such that $\log K$ should be close to the bits-back bitrate $D_{KL}(q(Z_i|X)||\mathcal{N}(0, 1))$. In practice, setting $\log_2 K = \lceil D_{KL(2)}(q(Z_i|X)||\mathcal{N}(0, 1)) \rceil$ typically yields a small enough reconstruction error, where $\lceil \cdot \rceil$ denotes the rounding operator. Using a larger $K$ does not provide additional benefits, while a smaller $K$ increases the error significantly.

### 3.3 PRACTICAL IMPLEMENTATION WITH TARGET DIVERGENCE CONSTRAINT

There are two challenges to make GQ practical. The first challenge is, if we want to construct a VQ-VAE with a specific codebook size $K$, how can we train a Gaussian VAE with corresponding KL divergence? The second challenge is, for a vanilla Gaussian VAE trained to minimize the loss in Eq.2, the values of $D_{KL(2)}(q(Z_i|X)||\mathcal{N}(0,1))$ vary significantly across dimensions $i$. How to train a Gaussian VAE with KL divergence remains close to $\log K$ across each dimension?

To address these two challenges, we propose the **Target Divergence Constraint (TDC)**. TDC is designed to ensure that $D_{KL(2)}(q(Z_i|X)||\mathcal{N}(0,1))$ is close to $\log_2 K$ for all dimensions $i = 1, \ldots, d$. Specifically, we set the target KL divergence as $\log_2 K$. For each dimension $i$, we impose a greater penalty if the KL divergence exceeds $\log_2 K + \alpha$ bits, and a smaller penalty if it falls below $\log_2 K - \alpha$ bits by using different $\lambda$s for each case, where $\alpha$ is the hyper-parameter controlling thresholding:

$$\mathcal{L}_{TDC} = \sum_{i=1}^{d} \lambda_i D_{KL}(q(Z_i|X)||\mathcal{N}(0,1)) + \Delta(X, g(z)),$$

$$\text{where } \lambda_i = \begin{cases} \lambda_{\min}, & D_{KL(2)}(q(Z_i|X)||\mathcal{N}(0,1)) < \log_2 K - \alpha \text{ bits}, \\ \lambda_{\text{mean}}, & D_{KL(2)}(q(Z_i|X)||\mathcal{N}(0,1)) \in [\log_2 K - \alpha, \log_2 K + \alpha] \text{ bits}, \\ \lambda_{\max}, & D_{KL(2)}(q(Z_i|X)||\mathcal{N}(0,1)) > \log_2 K + \alpha \text{ bits}. \end{cases} \quad (6)$$

To determine $\lambda_{\min}, \lambda_{\text{mean}}, \lambda_{\max}$, we extend the heuristic in MIRACLE (Havasi et al., 2018b) and HiFiC (Mentzer et al., 2020). More specifically, we initialize $\lambda_{\min} = \lambda_{\text{mean}} = \lambda_{\max} = 1$, and update them according to the following rule:

$$\lambda_{\min} = \lambda_{\min} \times \beta \text{ if } \min_i\{D_{KL(2)}(q(Z_i|X)||\mathcal{N}(0,1))\} > \log_2 K - \alpha \text{ else } \lambda_{\min}/\beta,$$

$$\lambda_{\text{mean}} = \lambda_{\text{mean}} \times \beta \text{ if } \text{mean}_i\{D_{KL(2)}(q(Z_i|X)||\mathcal{N}(0,1))\} > \log_2 K \text{ else } \lambda_{\text{mean}}/\beta,$$

$$\lambda_{\max} = \lambda_{\max} \times \beta \text{ if } \max_i\{D_{KL(2)}(q(Z_i|X)||\mathcal{N}(0,1))\} > \log_2 K + \alpha \text{ else } \lambda_{\max}/\beta, \quad (7)$$

where $\beta$ is the hyper-parameter controlling update speed. To avoid numerical issue, we further clip $\lambda_{\min}, \lambda_{\text{mean}}, \lambda_{\max}$ into a range of $[10^{-3}, 10^3]$ after each update. In practice, we use $\alpha = 0.5, \beta = 1.01$. In Appendix B, we propose an alternative implementation of TDC using Lambert W function, which is less effective for ViT models.

### 3.4 GROUPING TO MULTIPLE DIMENSIONS

The vanilla VQ-VAE has three key parameters: codebook size, codebook dimension and number of token. Some previous alternative to VQ-VAE, such as LFQ and BSQ (Yu et al., 2023; Zhao et al., 2024) limit the codebook dimension to 1. To support codebook dimension greater than 1, we can group $m$ tokens into a single large token with codebook size $\log_2 K = \left\lceil \sum_{l=i}^{i+m} D_{KL}(q(Z_l|X)||\mathcal{N}(0,1)) \right\rceil$. There are three grouping strategies to achieve this: post-quantization (PQ), post-training (PT), and training-based (TR), with different trade-offs in flexibility and performance. We brief these strategies in main text and provide details in Appendix C.2.

PQ is the most flexible grouping strategy and can be applied after GQ. For PQ, since the posterior of the Gaussian VAE $q(Z|X)$ is a factorized Gaussian, the quantization is also independent across dimensions. This means that we can trivially combine $m$ tokens into a larger one, by treating each token as an integer in $K^{1/m}$-based number system and aggregate them, which is the same as other one dimensional VQ-VAEs (Chang et al., 2022; Mentzer et al., 2023; Zhao et al., 2024).

PT is less flexible than PQ, as it can only be applied before GQ and after training the Gaussian VAE. For PT, we can view the one-dimensional GQ in Eq. 3 as the maximum likelihood estimator of a one-dimensional Gaussian. This approach can be extended to an $m$-dimensional diagonal Gaussian distribution. Additionally, for low bitrate cases, we observe that $m$-dimensional PT leads to low codebook usage. This is because $|\mu_i|$ is bounded by $\sqrt{2D_{KL}}$. When the $D_{KL}$ is small, some vector in the codebook that is far from 0 is never used. To address this, we introduce a regularization term weighted by $\omega$ to improve codebook usage by encouraging the selection $c_j$ that is far from 0:

$$\hat{z}_{i:i+m} = \arg\min_{c_j \in \{c_{1:K}\}} ||(\mu_{i:i+m} - c_j)/\sigma_{i:i+m}|| - \omega||c_j||, \text{ where } c_{1:K} \sim \mathcal{N}(0, I_m). \quad (8)$$

TR is the least flexible strategy and must be used during the training of the Gaussian VAE. Specifically, building on the PT quantization approach, we can relax the TDC training target by considering the relationship between $\sum_{l=i}^{i+m} D_{KL}(q(Z_l|X)||\mathcal{N}(0,1))$ and $\log_2 K \pm 0.5$ bits. In terms of performance, PQ does not affect reconstruction at all. PT provides slight improvements in reconstruction, while TR significantly enhances reconstruction performance (see Table 13).

## 3.5 IMPROVING TOKENBRIDGE WITH TARGET DIVERGENCE CONSTRAINT

TokenBridge (Wang et al., 2025) also convert a pre-trained Gaussian VAE into a VQ-VAE. It adopts the Post Training Quantization (PTQ) technique from model compression and proposes to treat latent as model parameters to discretize. It uses a fixed codebook composed of $2^K$ centroids of a Gaussian distribution. It then quantizes the posterior sample by finding the closest centroid. However, TokenBridge directly quantizes a vanilla Gaussian VAE without limiting the KL divergence of each dimension. This leads to suboptimal rate-distortion performance. In fact, we can also improve the performance of TokenBridge using TDC. Specifically, the quantization centers of TokenBridge are the equal-probability partition centers of the $\mathcal{N}(0,1)$ distribution, which can be seen as a special case of GQ with an evenly distributed codebook $c_{1:K}$. From this perspective, the number of PTQ bits should also match the bits-back coding bitrate of the Gaussian VAE, and TDC can therefore enhance the performance of TokenBridge. As we demonstrate in Table 3, TDC indeed improves TokenBridge performance by a large margin.

## 3.6 RELATIONSHIP WITH REVERSE CHANNEL CODING

GQ is closely related to reverse channel coding, which aims to simulate a distribution $q$ using samples from a distribution $p$ (Harsha et al., 2007; Li & El Gamal, 2018; Havasi et al., 2018b; Flamich et al., 2020; Theis & Yosri, 2021; Flamich et al., 2022; He et al., 2024). For example, Minimal Random Coding (MRC) (Havasi et al., 2018b), when applied to a Gaussian VAE, samples from the categorical distribution with logits given by the likelihood difference:

$$\hat{z}_i \sim \hat{q}(c_{1:K}) \propto e^{\log q(Z_i=c_j|X) - \log \mathcal{N}(c_j|0,1)}. \tag{9}$$

The key difference between MRC and GQ is that MRC and its variants (Havasi et al., 2018b; Theis & Yosri, 2021; Flamich et al., 2022; He et al., 2024) simulate a distribution through stochastic sampling, whereas a VQ-VAE requires deterministic quantization. For one dimension quantization, the bias bound of MRC derived from Chatterjee & Diaconis (2015) can not be achieved as VQ-VAE does not allow stochastic encoding. On the other hand, our achievability and converse bound is compatible with deterministic quantization. In terms of quantization error, GQ outperforms MRC by definition (Eq. 3). Besides, GQ without grouping (m=1) can be implemented by bisect search with better asymptotic complexity (See Appendix D.11).

Additionally, TDC is closely related to the MIRACLE heuristic and IsoKL parametrization of Gaussian VAE (Havasi et al., 2018a; Flamich et al., 2022; Lin et al., 2023). More specifically, MIRACLE also proposes adjusting $\lambda$ during VAE training. However, MIRACLE only maintains a single $\lambda$, making it less effective for controlling the mean of $D_{KL}$ but less effective for constraining its minimum and maximum values. On the other hand, IsoKL imposes strict control on $D_{KL}$ by directly solving for $\sigma$ given $\mu$ using the Lambert W function (Corless et al., 1996; Brezinski, 1996). However, IsoKL suffers from numerical issues and leads to suboptimal performance.

# 4 EXPERIMENTAL RESULTS

## 4.1 EXPERIMENTAL SETUP

**Models and Baselines** For image reconstruction, we select two representative autoencoder architectures: **UNet** from Stable Diffusion 3 (Esser et al., 2024), and **ViT** from the BSQ (Zhao et al., 2024). For VQ-VAE baselines, we include vanilla **VQGAN** (Van Den Oord et al., 2017) and several representative variants, including **FSQ** (Mentzer et al., 2023), **LFQ** (Yu et al., 2023), and **BSQ** (Zhao et al., 2024). Besides, we compare our approach to pre-trained VQ-VAE of **VQGAN-Taming** (Esser et al., 2021), **VQGAN-SD** (Rombach et al., 2022) **Llama-Gen** (Sun et al., 2024b), **FlowMo** (Sargent et al., 2025), **BSQ** (Zhao et al., 2024) and other conversion approaches such as **TokenBridge**

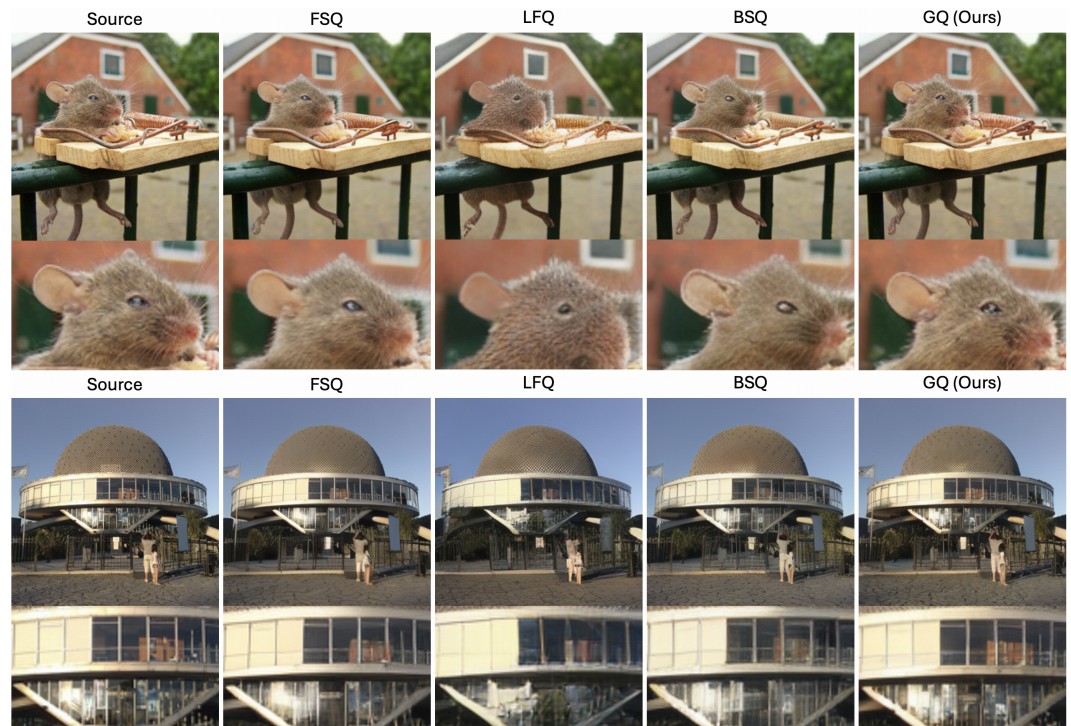

Figure 2: Qualitative results on ImageNet dataset and 0.25 bpp. Our GQ has most visually pleasing reconstruction result.

and **ReVQ** . Additionally, we demonstrate that our TDC technique can improve TokenBridge (Wang et al., 2025), a previous training-free approach for converting Gaussian VAEs into VQ-VAE. For image generation, we employ the Llama transformer (Touvron et al., 2023; Shi et al., 2024a).

**Datasets, Bitrates, and Metrics** For the datasets, we use the **ImageNet** (Deng et al., 2009) training split for training, and both the **ImageNet** and **COCO** (Lin et al., 2014) validation split for testing. For reconstruction and generation experiments, all images are resized to $256 \times 256$ and $128 \times 128$ respectively. In terms of bitrates, we evaluate the image reconstruction performance of all models using a **codebook size** of $2^{14} - 2^{18}$ and **token numbers** of 1024, 2048, and 4096, which correspond to bpp (bits-per-pixel) values of $0.22 - 1.00$. This extends the BSQ evaluation beyond $0.25 - 0.50$ bpp. For metrics, we adopt Peak Signal-to-Noise Ratio (**PSNR**), Learned Perceptual Image Patch Similarity (**LPIPS**) (Zhang et al., 2018), Structural Similarity Index Measure (**SSIM**) (Wang et al., 2004), and reconstruction Fréchet Inception Distance (**rFID**) (Heusel et al., 2017) for image reconstruction; generation Fréchet Inception Distance (**gFID**) and Inception Score (**IS**) (Salimans et al., 2016) are used for image generation. For further details, see Appendix C.

### 4.2 MAIN RESULTS

**Image Reconstruction** In Table 1 and Table 15, we compare our GQ method to other quantization approaches across the 0.25–1.00 bpp range. The results show that, in terms of reconstruction metrics such as PSNR, LPIPS, SSIM, and rFID, our GQ approach achieves state-of-the-art performance in most cases. The advantage of GQ is consistent across both UNet and ViT model architectures, as well as for both the ImageNet and COCO datasets. Visually, in Figure 2, it is shown that our GQ also produces pleasing reconstruction and preserves a lot more details in the source image. Besides, in Table. 2, we show that our GQ achieves competitive performance compared with several pre-trained models such as FlowMo, with less training epochs. Additionally, in Table. 11, we show that our GQ outperforms previous methods to discretize Gaussian VAE, including TokenBridge and ReVQ.

**Improving TokenBridge** In Table 3, we compare TokenBridge (Wang et al., 2025) applied to a vanilla Gaussian VAE and a TDC constrained Gaussian VAE. The results show that the quantization

Table 1: Quantitive results on ImageNet dataset. Our GQ outperforms other VQ-VAEs on both UNet and ViT architecture, across 0.25-1.00 bpp. **Bold**: best.

| Method | bpp (# of tokens) | UNet based | | | | ViT based | | | |
|---|---|---|---|---|---|---|---|---|---|
| | | PSNR↑ | LPIPS↓ | SSIM↑ | rFID↓ | PSNR↑ | LPIPS↓ | SSIM↑ | rFID↓ |
| VQGAN | | 26.51 | 0.125 | 0.748 | 5.714 | 25.39 | 0.103 | 0.740 | 3.518 |
| FSQ | 0.25 | 26.34 | 0.075 | 0.756 | 1.125 | 25.87 | 0.109 | 0.751 | 3.856 |
| LFQ | $(2^{16} \times 1024)$ | 24.74 | 0.164 | 0.722 | 16.337 | 24.81 | 0.143 | 0.725 | 15.716 |
| BSQ | | 25.62 | 0.086 | 0.754 | 1.080 | 26.52 | 0.083 | 0.793 | 1.649 |
| GQ (Ours) | | **27.61** | **0.059** | **0.807** | **0.529** | **27.88** | **0.061** | **0.823** | **0.932** |
| VQGAN | | 29.21 | 0.052 | 0.831 | 1.600 | 27.86 | 0.062 | 0.823 | 1.228 |
| FSQ | 0.50 | 29.29 | 0.047 | 0.845 | 0.871 | 28.83 | 0.055 | 0.842 | 1.067 |
| LFQ | $(2^{16} \times 2048)$ | 26.90 | 0.107 | 0.800 | 8.035 | 27.87 | 0.068 | 0.829 | 2.444 |
| BSQ | | 27.88 | 0.059 | 0.836 | 0.788 | 28.44 | 0.051 | 0.852 | 0.700 |
| GQ (Ours) | | **30.17** | **0.039** | **0.875** | **0.492** | **30.42** | **0.037** | **0.882** | **0.592** |
| VQGAN | | 32.06 | 0.026 | 0.896 | 0.580 | 31.32 | 0.032 | 0.899 | 0.716 |
| FSQ | 1.00 | 32.38 | 0.025 | 0.905 | 0.636 | 31.58 | 0.026 | 0.905 | 0.544 |
| LFQ | $(2^{16} \times 4096)$ | 28.31 | 0.074 | 0.840 | 3.617 | 26.67 | 0.105 | 0.790 | 8.288 |
| BSQ | | 30.50 | 0.032 | 0.900 | 0.346 | 31.60 | 0.027 | **0.914** | 0.379 |
| GQ (Ours) | | **32.47** | **0.023** | **0.907** | **0.322** | **31.71** | **0.024** | 0.903 | **0.349** |

Table 2: Quantitive results on ImageNet dataset. Our GQ outperforms previous pre-trained VQ-VAEs with less training. **Bold**: best, [*]: from paper, -: not available.

| Method | bpp (# of tokens) | PSNR↑ | LPIPS↓ | SSIM↑ | rFID↓ | ImageNet Training Epochs↓ | Params(M)↓ |
|---|---|---|---|---|---|---|---|
| VQGAN-Taming[*] | | 23.38 | - | - | 1.190 | (OpenImages) | 67 |
| VQGAN-SD[*] | 0.22 | - | - | - | 1.140 | (OpenImages) | 83 |
| Llama-Gen-32[*] | $(2^{14} \times 1024)$ | 24.44 | **0.064** | 0.768 | 0.590 | 40 | 70 |
| FlowMo-Hi[*] | | 24.93 | 0.073 | **0.785** | 0.560 | 300 | 945 |
| GQ (Ours) | | **25.31** | **0.064** | 0.762 | **0.491** | 40 | 82 |
| BSQ[*] | 0.28 | 27.78 | 0.063 | **0.817** | 0.990 | 200 | 175 |
| GQ (Ours) | $(2^{18} \times 1024)$ | **27.86** | **0.054** | 0.804 | **0.424** | 40 | 87 |

error of TokenBridge is quite large when applied to a vanilla Gaussian VAE. In contrast, TDC significantly reduces the quantization error.

**Image Generation** In Table 4, we evaluate the performance of GQ in terms of image generation. It is shown that compared with VQGAN, FSQ, LFQ and BSQ, our GQ has higher codebook usage and codebook entropy. In terms of generation FID and IS, our GQ is comparable to FSQ and better than other methods. Additionally, we train a DiT (Peebles & Xie, 2022) with same model architecture and same training setting, using the Gaussian VAE with and without TDC. It is shown that for limited computation, auto-regressive generation is more efficient than diffusion generation, in terms of both FID and IS. This result shows that the conversion from Gaussian VAE to VQ-VAE facilities auto-regressive generation, which improves the efficiency of image generation.

**Complexity** Compared with Gaussian VAE, the overhead of GQ is negligible (See Appendix D.10).

Table 3: Quantitive results of improving TokenBridge based on UNet architecture. Our TDC improves TokenBridge significantly.

| Method | bpp (# of tokens) | ImageNet validation | | | | COCO validation | | | |
|---|---|---|---|---|---|---|---|---|---|
| | | PSNR↑ | LPIPS↓ | SSIM↑ | rFID↓ | PSNR↑ | LPIPS↓ | SSIM↑ | rFID↓ |
| Gaussian VAE | $\approx 1.00$ (-) | 32.73 | 0.022 | 0.910 | 0.490 | 32.64 | 0.018 | 0.917 | 2.380 |
| Gaussian VAE (w/ TDC) | | 32.61 | 0.023 | 0.906 | 0.460 | 32.69 | 0.019 | 0.919 | 2.717 |
| TokenBridge | | 28.24 | 0.045 | 0.869 | 0.823 | 28.19 | 0.043 | 0.878 | 4.167 |
| TokenBridge (w/ TDC) | 1.00 $(2^{16} \times 4096)$ | 31.67 | 0.025 | 0.903 | 0.385 | 31.56 | 0.022 | 0.910 | 2.171 |
| GQ (Ours) | | 32.60 | 0.022 | 0.908 | 0.280 | 32.53 | 0.020 | 0.917 | 2.153 |

Table 4: Quantitive results on class conditional image generation of ImageNet dataset. Our GQ achieves best codebook usage and competitive generation performance. **Bold**: best.

| Method | Codebook Usage↑ | Codebook Entropy ↑ | generation FID ↓ | IS ↑ |
|---|---|---|---|---|
| *Diffusion* | | | | |
| Gaussian VAE w/o TDC | - | - | 8.35 | 202.19 |
| Gaussian VAE w/ TDC | - | - | 8.47 | 205.94 |
| *Auto-regressive* | | | | |
| VQGAN | 16.4% | 4.36 | 8.01 | 151.40 |
| FSQ | 94.3% | 14.74 | **7.33** | 224.88 |
| LFQ | 24.9% | 9.65 | 7.73 | 142.09 |
| BSQ | 99.8% | 14.93 | 7.82 | 221.64 |
| TokenBridge | 94.6% | 14.94 | 7.82 | 198.24 |
| GQ (Ours) | **100.0%** | **15.17** | 7.67 | **230.79** |

## 4.3 ABLATION STUDIES

**Effectiveness of Pre-trained Gaussian VAE** It is possible to train a vanilla VQ-VAE (Van Den Oord et al., 2017) using the same codebook as GQ, which is equivalent to a VQ-VAE with a fixed, Gaussian noise codebook. It is also possible to directly train the Gaussian VAE neural network with the GQ target in Eq. 8 using Gumbel-Softmax (Jang et al., 2016; Maddison et al., 2016). However, as shown in Table 5, both methods do not converge well. Furthermore, fine-tuning GQ after initializing it with a pre-trained Gaussian VAE also has only marginal effect on performance. These results indicate that GQ's conversion from a pre-trained Gaussian VAE is necessary and sufficient.

Table 5: The effect of pre-trained Gaussian VAE. Converting GQ from pre-trained Gaussian VAE is better than training GQ from scratch.

| Method | Training target | bpp | PSNR↑ | LPIPS↓ | SSIM↑ | rFID↓ |
|---|---|---|---|---|---|---|
| GQ (from scratch) | Vanilla VQ-VAE | | 8.50 | 0.763 | 0.156 | 360.597 |
| GQ (from scratch) | Gumbel-softmax Eq. 8 | 1.00 | 29.65 | 0.044 | 0.866 | 0.928 |
| GQ (finetune from Gaussian VAE) | Gumbel-softmax Eq. 8 | | 32.45 | 0.022 | 0.905 | 0.264 |
| GQ (convert from Gaussian VAE) | no | | 32.47 | 0.023 | 0.907 | 0.327 |

**Effectiveness and Alternatives of TDC** To demonstrate the necessity of TDC in Eq. 6, we train a vanilla Gaussian VAE without TDC. As shown in Table 6, the mean $D_{KL(2)}$ of the vanilla Gaussian VAE is close to that of the Gaussian VAE with TDC (3.99 vs. 4.26 bits). However, the range of $D_{KL(2)}$ is much wider for the vanilla model (0.26–27.29 vs. 2.93–5.63 bits). Although the reconstruction performance of the two Gaussian VAEs is very similar (PSNR: 32.73 vs. 32.61 dB, rFID: 0.490 vs. 0.460), GQ with TDC outperforms GQ without TDC by a large margin (PSNR: 31.25 vs. 26.43 dB, rFID: 0.372 vs. 0.978). This demonstrates that TDC is necessary.

Alternatives to TDC are the MIRACLE heuristic (Havasi et al., 2018a) and IsoKL (Flamich et al., 2022). MIRACLE is not that effective in terms of controlling the range is outperformed by TDC (PSNR 29.48 vs. 32.11 dB). On the other hand, IsoKL imposes a stricter constraint by requiring that $D_{KL}$ is exactly the same across all dimensions. Iso-KL enforece the constraint well but has inferior performance (PSNR 30.45 vs. 32.11 dB). This is because IsoKL is not numerically stable and it discards the solution with $\sigma_i^2 > 1$. In Appendix B, we propose a numerically stable version of Mean-KL (Lin et al., 2023), which is a IsoKL that supports grouping ($m > 1$). However, it does not work well for ViT based models.

**Alternatives to GQ** There are several stochastic alternatives to GQ, such as MRC, ORC, and A* coding (Havasi et al., 2018b; Theis & Yosri, 2021; Flamich et al., 2022; He et al., 2024). In Table 7, we compare these methods in terms of reconstruction quality. When applied to a TDC-constrained Gaussian VAE, GQ achieves the best PSNR, SSIM, and rFID. Besides, when grouping $m = 1$, GQ can be implemented using bisect search. This means that GQ is asymptotically faster than MRC, ORC and A* coding (See Appendix D.11).

**The TDC Parameters and Grouping Strategies** See Appendix D.

Table 6: The effect of adding constraints to Gaussian VAE. GQ is effective only on Gaussian VAE trained with TDC constraint.

| Methods | Constraint | $D_{KL(2)}$ mean, min-max | $\log_2 K$ | bpp | PSNR↑ | LPIPS↓ | SSIM↑ | rFID↓ |
|---|---|---|---|---|---|---|---|---|
| Gaussian VAE | no | 3.99, 0.26-27.29 | - | 1.00 | 32.73 | 0.022 | 0.910 | 0.490 |
| GQ | - | - | 4 | 1.00 | 26.43 | 0.834 | 0.054 | 0.978 |
| Gaussian VAE | MIRACLE | 4.34,0.91-26.98 | - | 1.00 | 32.82 | 0.023 | 0.910 | 0.436 |
| GQ | - | - | 4 | 1.00 | 29.48 | 0.039 | 0.887 | 0.439 |
| Gaussian VAE | IsoKL | 4.34, 4.24-4.38 | - | 1.00 | 30.54 | 0.878 | 0.027 | 0.400 |
| GQ | - | - | 4 | 1.00 | 30.45 | 0.878 | 0.030 | 0.468 |
| Gaussian VAE | TDC (ours) | 4.26, 2.93-5.63 | - | 1.06 | 32.61 | 0.023 | 0.906 | 0.460 |
| GQ | - | - | 4 | 1.00 | 32.11 | 0.023 | 0.906 | 0.414 |

Table 7: Comparison between MRC methods and GQ on ImageNet dataset. GQ has better reconstruction quality and can be implemented asymptotically faster when grouping $m = 1$.

| Methods | Encoding / Decoding Complexity | PSNR↑ | LPIPS↓ | SSIM↑ | rFID↓ |
|---|---|---|---|---|---|
| Gaussian VAE (w/ TDC) | $O(1)/O(1)$ | 32.75 | 0.023 | 0.906 | 0.460 |
| MRC (original) | $O(2^{DKL(q(Z_i\|X)\|\mathcal{N}(0,1))})/O(1)$ | 32.09 | 0.023 | 0.906 | 0.425 |
| MRC (A* coding) | $O(D_\infty(q(Z_i\|X)\|\mathcal{N}(0,1)))/O(D_\infty(q(Z_i\|X)\|\mathcal{N}(0,1)))$ | 32.09 | 0.023 | 0.906 | 0.425 |
| ORC | $O(2^{DKL(q(Z_i\|X)\|\mathcal{N}(0,1))})/O(1)$ | 32.09 | 0.023 | 0.906 | 0.419 |
| GQ (Ours) | $O(D_{KL}(q(Z_i\|X)\|\mathcal{N}(0,1)))/O(1)$ | 32.11 | 0.023 | 0.907 | 0.414 |

## 5 RELATED WORKS

VQ-VAE (Van Den Oord et al., 2017) is an autoencoder that compresses images into discrete tokens. Due to the discretization, it is impossible to be train directly using gradient descent. Various techniques have been proposed to address this, such as commitment loss (Van Den Oord et al., 2017), expectation maximization (EM) (Roy et al., 2018), the straight-through estimator (STE) (Bengio et al., 2013), and Gumbel-softmax (Jang et al., 2016; Maddison et al., 2016; Sønderby et al., 2017; Shi et al., 2024b). In addition, VQ-VAE is prone to codebook collapse. To mitigate this, various methods have been proposed, such as reducing the code dimension (Yu et al., 2021a; Sun et al., 2024a), product quantization (Zheng et al., 2022), residual quantization (Lee et al., 2022), dynamic quantization (Huang et al., 2023), multi-level quantization (Razavi et al., 2019), feature projection (Zhu et al., 2024), rotation codebooks (Fifty et al., 2024) and *etc.* (Yu et al., 2021b; Chiu et al., 2022; Takida et al., 2022; Zhang et al., 2023; Huh et al., 2023; Gautam et al., 2023; Goswami et al., 2024).

More related to our work, some variants of VQ-VAE with fixed codebook emerge, such as FSQ (Mentzer et al., 2023), LFQ (Yu et al., 2023), and BSQ (Zhao et al., 2024). However, training tricks such as the straight-through estimator (STE) is still required. Among all previous works, TokenBridge (Wang et al., 2025) and ReVQ (Zhang et al., 2025) are most aligned with our objective. TokenBridge and ReVQ also convert a pre-trained Gaussian VAE into a VQ-VAE. However, TokenBridge and ReVQ do not constraint the divergence of Gaussian VAE, leading to suboptimal performance. Besides, ReVQ requires some training while our approach is training-free.

**Reverse Channel Coding** See Section. 3.6.

## 6 CONCLUSION & DISCUSSION

In this paper, we propose **Gaussian Quant (GQ)**, an extremely simple yet effective technique that converts a pre-trained Gaussian VAE into a VQ-VAE without any additional training. Theoretically, we show that when the logarithm of the GQ codebook size exceeds the bits-back coding bitrate of the Gaussian VAE, a small quantization error is achieved. In addition, we propose a target divergence constraint (TDC) to implement GQ effectively. Empirically, we demonstrate that GQ outperforms previous discrete VAEs, such as VQGAN, FSQ, LFQ, and BSQ (Van Den Oord et al., 2017; Mentzer et al., 2023; Yu et al., 2023; Zhao et al., 2024). Furthermore, our TDC also improves the performance of TokenBridge (Wang et al., 2025).

We limit our evaluation of GQ to the standard SD3.5 UNet (Esser et al., 2024) and the BSQ-ViT architecture. Additionally, we restrict the bpp range to 0.22–1.00, which extends the BSQ's original range of 0.25–0.50 bpp. We acknowledge that there are several highly competitive VQ-VAEs that adopt multi-scale or residual architectures (Tian et al., 2024; Han et al., 2024) or study the low bpp regime (bpp $\leq 0.2$) (Yu et al., 2024; Sargent et al., 2025; Zhang et al., 2025). However, in this paper, we use standard architectures and a typical bpp range to focus on the core aspects of the quantization method. Additionally, we focus on achieving a good trade-off between bitrate and reconstruction quality, leaving the complex relationship between reconstruction and generation performance to future works (Wang et al., 2024; Xiong et al., 2025; Hansen-Estruch et al., 2025).

## ETHICS STATEMENT

The approach proposed in this paper focus on reconstruction of existing images with limited bitrate. As the model is essentially not generative, the ethic concerns is not obvious. Nevertheless, the GAN module in decoder might has negative effects, including issues related to mis-representation and trustworthiness.

## REPRODUCIBILITY STATEMENT

For theoretical results, the proofs for all theorems are presented in Appendix A. For the experiments, we use publicly accessible datasets such as ImageNet (Deng et al., 2009). Implementation details and hyper-parameters are provided in Appendix C. Besides, we include the source code for reproducing the experimental results in the supplementary material.

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

## A   PROOF OF MAIN RESULTS

**Theorem 1.** *Denote the mean and standard deviation of $q(Z_i|X = x)$ as $\mu_i, \sigma_i$, and assuming $|\mu_i|, \sigma_i$ 's product and sum are bounded by $|\mu_i\sigma_j| \leq c_1$, $|\mu_i| + |\sigma_j| \leq c_2$, then the probability of quantization error $|\hat{z}_i - \mu_i| \geq \sigma_i$ decays double exponentially to the amount of nats $t$ that codebook bitrate $\log K$ exceeds the bits-back coding bitrate, i.e.,*

$$\text{when } \log K = D_{KL}(q(Z_i|X)||\mathcal{N}(0,1)) + t,$$

$$\Pr\{|\hat{z}_i - \mu_i| \geq \sigma_i\} \leq \exp\left(-e^t \underbrace{\sqrt{\frac{2}{\pi}}e^{-c_1-0.5}}_{\text{constant}}\right). \tag{6}$$

*Proof.* Denote the cumulative distribution function (CDF) of $\mathcal{N}(0,1)$ as $\Phi$, and probability density function (PDF) of $\mathcal{N}(0,1)$ as $\phi$, then we need to consider the probability that no samples falls between $[\mu_i - \sigma_i, \mu_i + \sigma_i]$, which is

$$\Pr\{|\hat{z}_i - \mu_i| \geq \sigma_i\} = (1 - (\Phi(\mu_i + \sigma_i) - \Phi(\mu_i - \sigma_i)))^K. \tag{10}$$

Now we use the Bernoulli inequality, that $\forall y \in \mathbb{R}, 1 + y \leq e^y$. Let $y = -(\Phi(\mu_i + \sigma_i) - \Phi(\mu_i - \sigma_i))$, we have

$$1 - (\Phi(\mu_i + \sigma_i) - \Phi(\mu_i - \sigma_i)) \leq \exp\left(-(\Phi(\mu_i + \sigma_i) - \Phi(\mu_i - \sigma_i))\right). \tag{11}$$

Taking Eq. 11 into Eq. 10, we have

$$\Pr\{|\hat{z}_i - \mu_i| \geq \sigma_i\} \leq \exp\left(-(\Phi(\mu_i + \sigma_i) - \Phi(\mu_i - \sigma_i))\right)^K \tag{12}$$
$$= \exp\left(-K \cdot (\Phi(\mu_i + \sigma_i) - \Phi(\mu_i - \sigma_i))\right)$$
$$= \exp\left(-K \cdot \int_{\mu_i-\sigma_i}^{\mu_i+\sigma_i} \phi(x)dx\right).$$

By integral mean value theorem, $\exists x' \in [\mu_i - \sigma_i, \mu_i + \sigma_i]$, such that

$$\int_{\mu_i-\sigma_i}^{\mu_i+\sigma_i} \phi(x)dx = 2\sigma_i\phi(x'). \tag{13}$$

And then we have

$$\Pr\{|\hat{z}_i - \mu_i| \geq \sigma_i\} \leq \exp\left(-K \cdot 2\sigma_i\phi(x')\right). \tag{14}$$

Next, we consider three cases: $\mu_i - \sigma_i \geq 0$, $\mu_i + \sigma_i \leq 0$, and $\mu_i - \sigma_i \leq 0 \leq \mu_i + \sigma_i$.

First, consider the case when $\mu_i - \sigma_i \geq 0$. Obviously we have $\phi(\mu_i + \sigma_i) \leq \phi(x')$, and we have

$$\Pr\{|\hat{z}_i - \mu_i| \geq \sigma_i\} \leq \exp\left(-K \cdot 2\sigma_i\phi(\mu_i + \sigma_i)\right) \tag{15}$$
$$= \exp\left(-K \cdot \sqrt{\frac{2}{\pi}}\sigma_i e^{-\frac{1}{2}(\mu_i+\sigma_i)^2}\right)$$
$$= \exp\left(-K \cdot \sqrt{\frac{2}{\pi}}e^{-\frac{1}{2}(\mu_i^2+\sigma_i^2-\log\sigma^2-1.0+1.0)-\mu_i\sigma_i}\right)$$
$$= \exp\left(-K \cdot \sqrt{\frac{2}{\pi}}e^{-D_{KL}(q(Z_i|X)||\mathcal{N}(0,1))-\mu_i\sigma_i-0.5}\right).$$

Notice that as $\mu_i - \sigma_i \geq 0, \sigma_i > 0$, we must have $\mu_i\sigma_i > 0$, then

$$\Pr\{|\hat{z}_i - \mu_i| \geq \sigma_i\} \leq \exp\left(-K \cdot \sqrt{\frac{2}{\pi}}e^{-D_{KL}(q(Z_i|X)||\mathcal{N}(0,1))-|\mu_{\max}\sigma_{\max}|-0.5}\right). \tag{16}$$

Similarly, we can show similar result for $\mu_i + \sigma_i \leq 0$. Obviously we have $\phi(\mu_i - \sigma_i) \leq \phi(x')$, and we have

$$
\begin{aligned}
\Pr\{|\hat{z}_i - \mu_i| \geq \sigma_i\} &\leq \exp\left(-K \cdot 2\sigma_i\phi(\mu_i - \sigma_i)\right) \\
&= \exp\left(-K \cdot \sqrt{\frac{2}{\pi}}\sigma_i e^{-\frac{1}{2}(\mu_i+\sigma_i)^2}\right) \\
&= \exp\left(-K \cdot \sqrt{\frac{2}{\pi}}e^{-\frac{1}{2}(\mu_i^2+\sigma_i^2-\log\sigma^2-1.0+1.0)+\mu_i\sigma_i}\right) \\
&= \exp\left(-K \cdot \sqrt{\frac{2}{\pi}}e^{-D_{KL}(q(Z_i|X)||\mathcal{N}(0,1))+\mu_i\sigma_i-0.5}\right) \\
&\leq \exp\left(-K \cdot \sqrt{\frac{2}{\pi}}e^{-D_{KL}(q(Z_i|X)||\mathcal{N}(0,1))-|\mu_{\max}\sigma_{\max}|-0.5}\right)
\end{aligned}
\tag{17}
$$

Now, consider the case when $\mu_i - \sigma_i < 0 < \mu_i + \sigma_i$, obviously we must have either $\phi(\mu_i - \sigma_i) \leq \phi(x')$, or $\phi(\mu_i + \sigma_i) \leq \phi(x')$. If $\phi(\mu_i + \sigma_i) \leq \phi(x')$, then the result is the same as $\mu - \sigma_i \geq 0$. If $\phi(\mu_i - \sigma_i) \leq \phi(x')$, then the result is the same as $\mu + \sigma_i \leq 0$.

Therefore, for all $\mu_i, \sigma_i$, we have

$$
\Pr\{|\hat{z}_i - \mu_i| \geq \sigma_i\} \leq \exp\left(-K \cdot \sqrt{\frac{2}{\pi}}e^{-D_{KL}(q(Z_i|X)||\mathcal{N}(0,1))-|\mu_{\max}\sigma_{\max}|-0.5}\right).
\tag{18}
$$

Taking the value of $K$ in, we have the result

$$
\begin{aligned}
\Pr\{|\hat{z}_i - \mu_i| \geq \sigma_i\} &\leq \exp\left(-K \cdot \sqrt{\frac{2}{\pi}}e^{-D_{KL}(q(Z_i|X)||\mathcal{N}(0,1))-|\mu_{\max}\sigma_{\max}|-0.5}\right) \\
&= \exp\left(-e^t \cdot \sqrt{\frac{2}{\pi}}e^{-|\mu_{\max}\sigma_{\max}|-0.5}\right) \\
&= \exp\left(-e^t \cdot \sqrt{\frac{2}{\pi}}e^{-c_1-0.5}\right).
\end{aligned}
\tag{19}
$$

$\square$

**Theorem 2.** *The probability of quantization error $|\hat{z}_i - \mu_i| \geq \sigma_i$ increase exponentially to the amount of nats $t$ that codebook bitrate $\log K$ lower than the bits-back coding bitrate, i.e.,*

$$
\text{when } \log K = D_{KL}(q(Z_i|X)||\mathcal{N}(0,1)) - t,
$$

$$
\Pr\{|\hat{z}_i - \mu_i| \geq \sigma_i\} \geq 1 - e^{-t}\underbrace{\sqrt{\frac{2}{\pi}}e^{0.5c_2^2-0.5}}_{\text{constant}}.
\tag{7}
$$

*Proof.* Similar to the proof of Theorem. 1, we have

$$
\Pr(|\hat{z}_i - \mu_i| \geq \sigma_i) = (1 - (\Phi(\mu_i + \sigma_i) - \Phi(\mu_i - \sigma_i)))^K.
\tag{20}
$$

Now we use an inequality, that $\forall y \in (0,1), K \in \mathbb{N}, K \geq 1, (1-y)^K \geq 1 - Ky$. This is due to the fact that $(1-y)^K$ is convex in $(0,1)$, and $1 - Ky$ is tangent line at $y = 0$. With this inequality, we have

$$
(1 - (\Phi(\mu_i + \sigma_i) - \Phi(\mu_i - \sigma_i)))^K \geq 1 - K(\Phi(\mu_i + \sigma_i) - \Phi(\mu_i - \sigma_i)).
\tag{21}
$$

Again, we can use integral mean value theorem, and find out that when $\mu_i - \sigma_i \geq 0$,

$$
\begin{aligned}
\Pr(|\hat{z}_i - \mu_i| \geq \sigma_i) &\geq 1 - K(\Phi(\mu_i + \sigma_i) - \Phi(\mu_i - \sigma_i)) \\
&\geq 1 - K 2\sigma_i \phi(\mu_i - \sigma_i) \\
&= 1 - K\sqrt{\frac{2}{\pi}}\sigma_i e^{-\frac{1}{2}(x_i - \sigma_i)^2} \\
&= 1 - K\sqrt{\frac{2}{\pi}}e^{-\frac{1}{2}(x_i^2 + \sigma_i^2 - \log\sigma_i^2 - 1.0) + |\mu_i\sigma_i| - 0.5} \\
&= 1 - K\sqrt{\frac{2}{\pi}}e^{-D_{KL}(q(Z_i|X)||\mathcal{N}(0,1)) + |\mu_i\sigma_i| - 0.5} \\
&\geq 1 - K e^{-D_{KL}(q(Z_i|X)||\mathcal{N}(0,1))}\sqrt{\frac{2}{\pi}}e^{0.5(\mu_i+\sigma_i)^2 - 0.5} \quad (22)
\end{aligned}
$$

Similar results can be obtained for $\mu_i + \sigma_i \leq 0$. For the case that $\mu_i - \sigma_i \leq 0 \leq \mu_i + \sigma_i$, we have

$$
\begin{aligned}
\Pr(|\hat{z}_i - \mu_i| \geq \sigma_i) &\geq 1 - K(\Phi(\mu_i + \sigma_i) - \Phi(\mu_i - \sigma_i)) \\
&\geq 1 - K 2\sigma_i \phi(0) \\
&= 1 - K\sqrt{\frac{2}{\pi}}\sigma_i e^{-\frac{1}{2}(0)^2} \\
&= 1 - K\sqrt{\frac{2}{\pi}}e^{-\frac{1}{2}(\mu_i^2 + \sigma_i^2 - \log\sigma_i^2 - 1.0) - 0.5 + 0.5(\mu_i^2 + \sigma_i^2)} \\
&= 1 - K\sqrt{\frac{2}{\pi}}e^{-D_{KL}(q(Z_i|X)||\mathcal{N}(0,1)) + 0.5(\mu_i^2 + \sigma_i^2) + |\mu_i\sigma_i| - 0.5} \\
&= 1 - K e^{-D_{KL}(q(Z_i|X)||\mathcal{N}(0,1))}\sqrt{\frac{2}{\pi}}e^{0.5(\mu_i+\sigma_i)^2 - 0.5} \quad (23)
\end{aligned}
$$

Taking the value of $K = e^{D_{KL}(q(Z_i||\mathcal{N}(0,1))) - t}$ , we have the result

$$
\begin{aligned}
\Pr(|\hat{z}_i - \mu_i| \geq \sigma_i) &\geq 1 - e^{D_{KL}(q(Z_i||\mathcal{N}(0,1))) - t - D_{KL}(q(Z_i|X)||\mathcal{N}(0,1))}\sqrt{\frac{2}{\pi}}e^{0.5(\mu_i+\sigma_i)^2 - 0.5} \\
&\geq 1 - e^{-t}\sqrt{\frac{2}{\pi}}e^{0.5c_2^2 - 0.5}. \quad (24)
\end{aligned}
$$

$\square$

To better illustrate the significance of these bounds, we provide a practical example. We evaluate the ImageNet validation dataset using a pre-trained Gaussian VAE and compute that $c_1 = 8.12$ and $c_2 = 1.50$. We then visualize the upper bound and lower bound of $\Pr|\hat{z} - \mu_i| \geq \sigma_i$ in Fig. 3. The results show that when the codebook bitrate exceeds the bits-back coding bitrate by approximately 10 nats, the probability of large quantization error diminishes to zero. Conversely, when the codebook bitrate is smaller than the bits-back coding bitrate, the probability of large quantization error increases.

## B    STABLE MEAN-KL PARAMETRIZATION

We investigate an alternative to TDC, namely the Mean-KL parametrization (Lin et al., 2023), which is considered to be easier to train than TDC since it does not require the construction of an empirical $R(\lambda)$ model.

### B.1    MEAN-KL PARAMETRIZATION

The Mean-KL parametrization (Lin et al., 2023) supports grouping with $m > 1$. Its neural network output consists of two $m$-dimensional tensors, $\hat{\gamma}i : i + m$ and $\tau i : i + m$, which allocate the $D_{\mathrm{KL}}$ target $K$ across the $m$ dimensions and determine the mean, respectively. More specifically, the

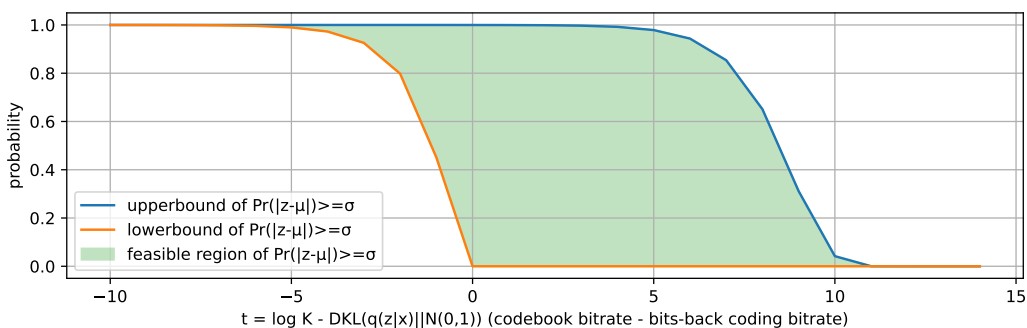

Figure 3: A visualization of large quantization error lowerbound and upperbound with ImageNet validation dataset.

Mean-KL parametrization determines the mean $\mu_{i:i+m}$ and variance $\sigma^2_{i:i+m}$ as follows, where $\mathcal{W}(\cdot)$ denotes the principal branch of the Lambert W function:

$$\gamma_{i:i+m} = \text{Softmax}(\hat{\gamma}_{i:i+m}),$$
$$\kappa_{i:i+m} = \gamma_{i:i+m}K,$$
$$\mu_{1:m} = \sqrt{2\kappa_{i:i+m}}\tanh(\tau_{i:i+m}),$$
$$\sigma^2_{i:i+m} = -\mathcal{W}(-\exp(\mu^2_{i:i+m} - 2\kappa_{i:i+m} - 1.0)). \tag{25}$$

The Mean-KL parametrization is designed for model compression. When directly applied to Gaussian VAEs, two typical cases may arise, as shown in Table 8, both of which can result in a not-a-number (NaN) error in floating-point computations.

Table 8: Two typical types of NaN in Mean-KL parametrization.

| $\mu_i$ | $\kappa_i$ | $\sigma^2_i$ |
|---|---|---|
| -2.7286 | 3.7227 | NaN |
| 0.0013 | $9.1458\times10^{-7}$ | NaN |

### B.2 STABLE MEAN-KL PARAMETRIZATION

It is evident that the two types of NaN errors are caused by excessively large values of $|\mu_i|$ and excessively small values of $\kappa_i$, respectively. To address this numerical issue, we propose the Stable Mean-KL parametrization, which introduces two regularization parameters, $r_1 = 0.1$ and $r_2 = 0.01$. The parameter $r_1$ ensures that each $\kappa_i \geq r_1/m$, while $r_2$ shrinks $\mu_i$ towards 0.

$$\kappa_{i:i+m} = \gamma_{i:i+m}(K - r_1) + r_1/m,$$
$$\mu_{1:m} = \sqrt{2\kappa_{i:i+m}}\tanh(\tau_{i:i+m})(1 - r_2), \tag{26}$$

### B.3 RESULTS OF STABLE MEAN-KL PARAMETRIZATION

In Table 9, we present the results of the Stable Mean-KL parametrization. For UNet-based models, Stable Mean-KL achieves performance comparable to TDC. However, for ViT-based models, Stable Mean-KL performs significantly worse than TDC. Since Stable Mean-KL does not consistently outperform TDC, we choose to use TDC for the final model. Nonetheless, if only UNet-based models are required, Stable Mean-KL can be an effective alternative to TDC, as it does not require an empirical $R(\lambda)$ model and is significantly simpler to train.

Table 9: Quantitive results on ImageNet validation dataset.

| Method | bpp (# of tokens) | UNet based | | | | ViT based | | | |
|---|---|---|---|---|---|---|---|---|---|
| | | PSNR↑ | LPIPS↓ | SSIM↑ | rFID↓ | PSNR↑ | LPIPS↓ | SSIM↑ | rFID↓ |
| GQ (Mean-KL) | 1.00 | NaN | NaN | NaN | NaN | NaN | NaN | NaN | NaN |
| GQ (Stable Mean-KL) | $(2^{16} \times 4096)$ | 32.35 | 0.023 | 0.905 | 0.280 | 30.80 | 0.030 | 0.891 | 0.556 |
| GQ (TDC) | | 32.47 | 0.023 | 0.907 | 0.322 | 31.71 | 0.024 | 0.903 | 0.349 |

# C  IMPLEMENTATION DETAILS

## C.1  DETAILS OF TRAINING AND DISTORTION OBJECTIVE

We train all VQ-VAEs on the ImageNet validation dataset using $8\times$ H100 GPUs for approximately 24 hours. For UNet models, we train each model for 30 epochs using the ADAM (Kingma & Ba, 2014) optimizer with a batch size of 128 and a learning rate of $1 \times 10^{-4}$. For ViT models, we train each model for 40 epochs using the ADAM optimizer with a batch size of 256 and a learning rate of $4 \times 10^{-7}$.

All VQ-VAEs are trained using the following distortion objective, which corresponds to the classical VQ-GAN (Esser et al., 2020) objective employed in the Stable Diffusion VAE (Rombach et al., 2022).

$$\Delta(X, g(z)) = \mathcal{L}_{MSE}(X, g(z)) + w_1 \mathcal{L}_{LPIPS}(X, g(z)) + w_2 \mathcal{L}_{GAN}(g(z)). \qquad (27)$$

Following the implementation of Stable Diffusion, we set $w_1 = 1.0$ and $w_2 = 0.75$ for UNet models. Consistent with the implementation of BSQ (Zhao et al., 2024), we set $w_1 = 0.1$ and $w_2 = 0.1$ for ViT models.

For the image generation model, we first train all VQ-VAEs using images of size $128 \times 128$, following the same settings as described above. Subsequently, we train the auto-regressive transformer for image generation using the implementation of IBQ (Shi et al., 2024b) with a Llama-base transformer architecture. The transformer has a vocabulary size of $2^{16}$, 16 layers, 16 attention heads, and an embedding dimension of 1024. We train the transformer for 100 epochs using the ADAM optimizer with a learning rate of $3 \times 10^{-4}$ and a batch size of 512.

## C.2  DETAILS OF GROUPING STRATEGIES

We extend the notation from main text. We group $m$ tokens into one large token with codebook size $K$. For each quantization output $\hat{z}_i$, we denote the corresponding index in the codebook as $\mathcal{I}_i$. And we have $c_{\mathcal{I}_i} = \hat{z}_i$.

**Post Quantization (PQ)** The Post Quantization (PQ) grouping strategy happens after GQ. We first train a Gaussian VAE with codebook size for each dimension $K^{1/m}$. Next, we obtain GQ tokens $\mathcal{I}_{1:d}$. Then, we group those $d$ tokens into $d/m$ groups with group size of $m$. Denote the group index as $g = 0, \ldots, d/m - 1$, then each group can be denoted as $\{\mathcal{I}_{gm+l}\}, l = 1, \ldots, m$.

In that case, we have $\max\{\mathcal{I}_{gm+l}\} \leq K^{1/m}$. Then, we can view each $\mathcal{I}_{gm+l}$ as an integer in a $K^{1/m}$-base numerical system. Then, aggregating $m$ tokens $\{\mathcal{I}_{gm+l}\}$ into one large token $I_g^m$ is as easy as concatenating $m$ tokens into a larger integer in a $K$-based numerical system: $\mathcal{I}_g^m = \sum_{l=1}^{m} \mathcal{I}_{gm+l} K^{(l-i)/m} \leq K$.

**Post Training (PT)** The Post Training (PT) grouping strategy happens after the training of Gaussian VAE. We still start with a Gaussian VAE with target codebook size for each dimension $K^{1/m}$. Next, instead of performing GQ for each dimension, we consider the following $m$ dimension GQ for each group $g$:

$$\hat{z}_{gm:gm+m} = \arg\min_{c_j \in \{c_{1:K}\}} ||(\mu_{gm:gm+m} - c_j)/\sigma_{gm:gm+m}||, \text{ where } c_{1:K} \sim \mathcal{N}(0, I_m). \qquad (28)$$

In fact, we can view one dimension GQ as a max likelihood:

$$\hat{z}_i = \arg\max_{c_j \in \{c_{1:K}\}} \log q(Z_i = c_j | X), \text{ where } c_{1:K} \sim \mathcal{N}(0, 1).$$

Then we can extend the max likelihood to $m$ dimension, which is equivalent to the basic version of PT in Eq. 29.

$$\hat{z}_{gm:gm+m} = \arg \max_{c_j \in \{c_{1:K}\}} \log q(Z_{gm:gm+m} = c_j | X), \text{ where } c_{1:K} \sim \mathcal{N}(0, I_m).$$

When group size $m$ is large, we observe that vanilla PT in Eq. 29 leads to codebook collapse (See Table. 13) and a decay in performance. Therefore, we include a regularization term weighted by $\omega$:

$$\hat{z}_{gm:gm+m} = \arg \min_{c_j \in \{c_{1:K}\}} ||(\mu_{gm:gm+m} - c_j)/\sigma_{gm:gm+m}|| - \omega ||c_j||, \text{ where } c_{1:K} \sim \mathcal{N}(0, I_m).$$

$$(29)$$

**Training Aware (TA)** The Training Aware (TR) grouping strategy happens before the training of Gaussian VAE. More specifically, we directly interoperate grouping with the TDC, and introduce the $m$ group TDC as follows:

$$\mathcal{L}^m_{TDC} = \sum_{g=0}^{d/m} \lambda_g \sum_{l=1}^{m} D_{KL}(q(Z_{gm+l}|X)||\mathcal{N}(0,1)) + \Delta(X, g(z)),$$

where $\lambda_g = \begin{cases} \lambda_{\min}, & \sum_{l=1}^{m} D_{KL(2)}(q(Z_{gm+l}|X)||\mathcal{N}(0,1)) < \log_2 K - \alpha \text{ bits}, \\ \lambda_{\text{mean}}, & \sum_{l=1}^{m} D_{KL(2)}(q(Z_{gm+l}|X)||\mathcal{N}(0,1)) \in [\log_2 K - 0.5, \log_2 K + 0.5] \text{ bits}, \\ \lambda_{\max}, & \sum_{l=1}^{m} D_{KL(2)}(q(Z_{gm+l}|X)||\mathcal{N}(0,1)) > \log_2 K + \alpha \text{ bits}. \end{cases}$

$$(30)$$

Besides, the $\lambda$ update heuristic should also consider $m$ dimension as a group:

$$\lambda_{\min} = \lambda_{\min} \times \beta \text{ if } \min_g \{\sum_{l=1}^{m} D_{KL(2)}(q(Z_{gm+l}|X)||\mathcal{N}(0,1))\} > \log_2 K - \alpha \text{ else } \lambda_{\min}/\beta,$$

$$\lambda_{\text{mean}} = \lambda_{\text{mean}} \times \beta \text{ if } \text{mean}_g \{\sum_{l=1}^{m} D_{KL(2)}(q(Z_{gm+l}|X)||\mathcal{N}(0,1))\} > \log_2 K \text{ else } \lambda_{\text{mean}}/\beta,$$

$$\lambda_{\max} = \lambda_{\max} \times \beta \text{ if } \max_g \{\sum_{l=1}^{m} D_{KL(2)}(q(Z_{gm+l}|X)||\mathcal{N}(0,1))\} > \log_2 K + \alpha \text{ else } \lambda_{\max}/\beta. \quad (31)$$

## C.3 DETAILS OF HYPER-PARAMETERS

Below, we describe the implementation details along with the definition of hyperparameters for each method. In Table 10, we list the values of these hyperparameters for different bits-per-pixel (bpp) settings.

**VQGAN** (Van Den Oord et al., 2017) We adopt the factorized codebook VQGAN variant following (Zheng et al., 2022). For each codebook, we use a codebook size of $K = 2^{16}$ and a group dimension of $m = 16$. The number of codebooks $n$ varies depending on the bitrate. Additionally, we use a codebook loss weight of $\lambda = 1.0$ and a commitment loss weight of $\zeta = 0.25$.

**FSQ** (Mentzer et al., 2023) The only parameter of FSQ is the codebook list $l$, which represents the quantization level for each integer value. We set each unit value to $2^4 = 16$, and populate $l$ with the appropriate number of 16s according to the desired bitrate.

**LFQ** (Yu et al., 2023) For LFQ at 0.25 bpp, we follow the original paper and split a large codebook of size $2^{16}$ into $n = 2$ smaller codebooks, each with $K = 2^8$ and a codebook dimension of $m = 8$. We use an entropy loss weight of $\lambda = 0.1$ and a commitment loss weight of $\zeta = 0.025$.

**BSQ** (Zhao et al., 2024) We fix the size of each BSQ codebook to $2^1$, with a group dimension of $m = 1$, and vary the number of codebooks $n$ according to the desired bitrate. For the entropy penalization parameter, we set $\lambda = 0.1$, following the official implementation.

**GQ** We use TR grouping with a fixed codebook size of $K = 2^{16}$. Each group has dimension $m$, and there are $n$ groups in total, such that $m \times n = 16$. The value of $m$ varies depending on the bitrate.

Table 10: Details of Hyper-parameter values.

| | bpp | Hyper-parameters |
|---|---|---|
| VQ | 0.25 | $K = 2^{16}, n = 1, m = 16, \lambda = 1.0, \zeta = 0.25$ |
| | 0.50 | $K = 2^{16}, n = 2, m = 16, \lambda = 1.0, \zeta = 0.25$ |
| | 1.00 | $K = 2^{16}, n = 4, m = 16, \lambda = 1.0, \zeta = 0.25$ |
| FSQ | 0.25 | $l = \{16, 16, 16, 16\}$ |
| | 0.50 | $l = \{16, 16, 16, 16, 16, 16, 16, 16\}$ |
| | 1.00 | $l = \{16, 16, 16, 16, 16, 16, 16, 16, 16, 16, 16, 16, 16, 16, 16, 16\}$ |
| LFQ | 0.25 | $K = 2^8, n = 2, m = 8, \lambda = 0.1, \zeta = 0.025$ |
| | 0.50 | $K = 2^8, n = 4, m = 8, \lambda = 0.1, \zeta = 0.025$ |
| | 1.00 | $K = 2^8, n = 8, m = 8, \lambda = 0.1, \zeta = 0.025$ |
| BSQ | 0.25 | $K = 2^1, n = 16, m = 1, \lambda = 0.1$ |
| | 0.50 | $K = 2^1, n = 32, m = 1, \lambda = 0.1$ |
| | 1.00 | $K = 2^1, n = 64, m = 1, \lambda = 0.1$ |
| GQ | 0.25 | $K = 2^{16}, n = 1, m = 16, \omega = 2.0$ |
| | 0.50 | $K = 2^{16}, n = 2, m = 8, \omega = 2.0$ |
| | 1.00 | $K = 2^{16}, n = 4, m = 4, \omega = 0.0$ |

# D ADDITIONAL QUANTITIVE RESULTS

## D.1 COMPARISON TO OTHER CONVERSION METHODS

In Table. 11, we compare our GQ to TokenBridge (Wang et al., 2025) and ReVQ (Zhang et al., 2025), which also convert a Gaussian VAE into a VQ-VAE. Those results are from their original paper. It is shown that GQ has best reconstruction metrics. Besides, only GQ has theoretical guarantee.

Table 11: Quantitive results on ImageNet dataset. **Bold**: best, *: from paper, -: not available.

| Method | Training free | Theoretical guarantee | bpp (# of tokens) | PSNR↑ | LPIPS↓ | SSIM↑ | rFID↓ |
|---|---|---|---|---|---|---|---|
| OpenMagViT-V2* | No | No | $0.07 \ (2^{18} \times 256)$ | 21.63 | **0.111** | 0.640 | 1.17 |
| TokenBridge* | **Yes** | No | $0.375 \ (2^6 \times 4096)$ | - | - | - | 1.11 |
| ReVQ-256T* | No | No | $0.07 \ (2^{18} \times 256)$ | 21.96 | 0.121 | 0.640 | 2.05 |
| GQ (Ours) | **Yes** | **Yes** | $0.07 \ (2^{18} \times 256)$ | **22.30** | **0.116** | **0.642** | **1.04** |

## D.2 THE TDC PARAMETERS

In Table 12, we show the effect of different TDC parameters. It is shown that $\beta = 1.001, 1.01, 1.1$ does not make a significant different on result. However, setting $\alpha > 0.5$ is harmful to performance.

Table 12: Ablation study on TDC parameters.

| $\alpha$ | $\beta$ | PSNR↑ | LPIPS↓ | SSIM↑ | rFID↓ |
|---|---|---|---|---|---|
| 0.5 | 1.01 | 27.61 | 0.059 | 0.807 | 0.529 |
| 0.1 | 1.01 | 27.56 | 0.058 | 0.812 | 0.551 |
| 1.0 | 1.01 | 27.61 | 0.063 | 0.811 | 0.701 |
| 0.5 | 1.1 | 27.63 | 0.060 | 0.809 | 0.534 |
| 0.5 | 1.001 | 27.48 | 0.058 | 0.804 | 0.510 |

## D.3 EFFECTIVENESS OF GROUPING STRATEGIES

In Table 13, we evaluate the effect of token grouping techniques. The scenario we consider involves grouping four 4-bit tokens into a single 16-bit token, which is a reasonable setting for autoregressive

generation. The results show that PQ has no effect on reconstruction performance, while PT provides some improvements in PSNR and SSIM. In contrast, TR, which involves training the Gaussian VAE with a grouping target, achieves the best reconstruction performance.

Table 13: Effects of token grouping on ImageNet dataset. TR strategy has best reconstruction performance for grouping $m = 4$.

| | Grouping | $\log_2 K$ | $D_{KL(2)}$ mean, min-max | bpp | PSNR↑ | LPIPS↓ | SSIM↑ | rFID↓ |
|---|---|---|---|---|---|---|---|---|
| Gaussian VAE (w/ TDC) | no (m=1) | - | 4.26, 2.93-5.63 | 1.06 | 32.61 | 0.023 | 0.906 | 0.460 |
| GQ | no (m=1) | 4 | - | 1.00 | 32.11 | 0.023 | 0.906 | 0.414 |
| GQ | PQ (m=4) | 16 | - | 1.00 | 32.11 | 0.023 | 0.906 | 0.414 |
| GQ | PT (m=4) | 16 | - | 1.00 | 32.15 | 0.023 | 0.907 | 0.428 |
| Gaussian VAE (w/ TDC) | TR (m=4) | - | 15.99, 14.81-17.54 | 1.00 | 32.62 | 0.023 | 0.909 | 0.331 |
| GQ | TR (m=4) | 16 | - | 1.00 | 32.47 | 0.023 | 0.907 | 0.322 |

Additionally, in Table 14, we show the effect of the regularization parameter $\omega$ for PT and TR. For high bitrates, such as $1.00$ bpp, regularization is not required; in other words, setting $\omega = 0.0$ yields the good enough codebook usage and rFID. For lower bitrates, such as $0.50$ bpp, $\omega = 0.0$ leads to codebook collapse, while $\omega = 2.0$ achieves the best codebook entropy and rFID.

Table 14: Ablation Study on regularization $\omega$.

| bpp | $\omega$ | Codebook Usage↑ | Codebook Entropy↑ | PSNR↑ | LPIPS↓ | SSIM↑ | rFID↓ |
|---|---|---|---|---|---|---|---|
| 0.50 | 0.0 | 99.3% | 14.96 | 30.00 | 0.044 | 0.873 | 0.783 |
| | 1.0 | 100.0% | 15.14 | 30.35 | 0.040 | 0.877 | 0.589 |
| | 2.0 | 100.0% | 15.22 | 30.17 | 0.039 | 0.875 | 0.492 |
| | 4.0 | 100.0% | 14.81 | 28.08 | 0.061 | 0.846 | 1.269 |
| 1.00 | 0.0 | 100.0% | 15.05 | 32.47 | 0.023 | 0.907 | 0.322 |
| | 1.0 | 100.0% | 15.05 | 32.47 | 0.023 | 0.907 | 0.327 |
| | 2.0 | 100.0% | 15.06 | 32.47 | 0.024 | 0.907 | 0.332 |
| | 4.0 | 100.0% | 15.07 | 32.44 | 0.024 | 0.907 | 0.343 |

Table 15: Quantitive results on COCO 2017 dataset. **Bold**: best.

| Method | bpp (# of tokens) | UNet based | | | | ViT based | | | |
|---|---|---|---|---|---|---|---|---|---|
| | | PSNR↑ | LPIPS↓ | SSIM↑ | rFID↓ | PSNR↑ | LPIPS↓ | SSIM↑ | rFID↓ |
| VQGAN | | 26.25 | 0.099 | 0.756 | 14.110 | 25.11 | 0.106 | 0.747 | 11.231 |
| FSQ | 0.25 | 26.01 | 0.072 | 0.767 | 5.451 | 25.85 | 0.112 | 0.765 | 11.213 |
| LFQ | $(2^{16} \times 1024)$ | 24.60 | 0.164 | 0.722 | 32.789 | 24.46 | 0.143 | 0.729 | 29.975 |
| BSQ | | 25.29 | 0.085 | 0.763 | 5.803 | 26.15 | 0.082 | 0.798 | 7.034 |
| GQ (Ours) | | **27.29** | **0.057** | **0.816** | **3.797** | **27.55** | **0.060** | **0.830** | **5.305** |
| VQGAN | | 29.06 | 0.049 | 0.839 | 6.616 | 27.83 | 0.058 | 0.832 | 5.461 |
| FSQ | 0.50 | 29.08 | 0.043 | 0.855 | 4.008 | 28.51 | 0.053 | 0.851 | 5.390 |
| LFQ | $(2^{16} \times 2048)$ | 26.47 | 0.103 | 0.805 | 17.508 | 27.54 | 0.067 | 0.833 | 8.700 |
| BSQ | | 27.58 | 0.057 | 0.844 | 4.465 | 28.19 | 0.049 | 0.858 | 4.587 |
| GQ (Ours) | | **30.14** | **0.037** | **0.877** | **3.116** | **30.18** | **0.034** | **0.887** | **3.616** |
| VQGAN | | 31.97 | 0.024 | 0.901 | 3.455 | 31.07 | 0.029 | 0.904 | 3.494 |
| FSQ | 1.00 | 32.30 | 0.022 | **0.917** | 2.797 | 31.48 | 0.023 | 0.911 | 3.045 |
| LFQ | $(2^{16} \times 4096)$ | 28.16 | 0.072 | 0.845 | 11.121 | 26.36 | 0.103 | 0.794 | 20.381 |
| BSQ | | 30.33 | 0.031 | 0.906 | 2.638 | 31.38 | 0.026 | **0.918** | 2.835 |
| GQ (Ours) | | **32.36** | **0.020** | 0.915 | **1.875** | **31.50** | **0.022** | 0.908 | **2.703** |

## D.4 THE QUANTIZATION ERROR IN PIXEL SPACE

Previously we examine the quantization error in latent space. We can further discuss the quantization error in pixel space given the decoder is smooth. More specifically, we have:

Table 16: The effect of quantization in pixel space.

| Latents | bits per latent | PSNR↑ | LPIPS↓ | SSIM↑ | rFID↓ |
|---|---|---|---|---|---|
| $\mu_i = \mathbb{E}[Z_i\|X]$ (posterior mean) | 16 bits | 32.92 | 0.020 | 0.913 | 0.46 |
| $z_i \sim q(Z_i\|X)$ (Gaussian sample) | $D_{KL}(q(Z_i\|X)\|\|\mathcal{N}(0,1)) = 4.26$ bits | 32.61 | 0.021 | 0.911 | 0.46 |
| $\hat{z}_i$ (GQ) | $\log_2 K = 4$ bits | 32.11 | 0.023 | 0.906 | 0.414 |

**Corollary 3.** *Following the setting in **Theorem 1**, and assuming the decoder $g(.)$ satisfy $|g(x_1) - g(x_2)| \leq c_3|x_1 - x_2|$, we have:*

$$Pr\{|g(\hat{z}_i) - g(\mu_i)| \geq c_3\sigma_i\} \leq \exp(-e^t\sqrt{\frac{2}{\pi}}e^{-c_1-0.5}). \tag{32}$$

*Proof.* As $|g(\hat{z}_i) - g(\mu_i)| \leq c_3|\hat{z}_i - \mu_i|$, we have $Pr\{|g(\hat{z}_i) - g(\mu_i)| \geq c_3\sigma_i\} \leq Pr\{c_3|\hat{z}_i - \mu_i| \geq c_3\sigma_i\} = Pr\{|\hat{z}_i - \mu_i| \geq \sigma_i\}$.

We can see that theoretically, the quantization error can be magnified by the Lipschitz constant $c_3$. However, we note that this is not a significant issue in practice. As shown in the Table 16, the actual loss of quality caused by GQ remains reasonable.

## D.5 ROBUSTNESS OF CODEBOOK TO RANDOM SEED

As the codebook is usually large enough ($2^{16}$) to compensate for the randomness. We provide additional in Table 17 showing that the random seed has little effect on reconstruction performance. We used three continuos random seeds without cherry-picking. It is clear that the performance of GQ is not affected by randomness.

Table 17: The effect of codebook randomness on the performance of GQ.

| Random Seed | PSNR↑ | LPIPS↓ | SSIM↑ | rFID↓ |
|---|---|---|---|---|
| 42 | 27.61 | 0.059 | 0.807 | 0.529 |
| 43 | 27.61 | 0.059 | 0.807 | 0.523 |
| 44 | 27.62 | 0.059 | 0.807 | 0.526 |

## D.6 THE EFFECT OF SIMPLY INCREASING CODEBOOK SIZE

According to Theorem 1, increasing $\log K$ too much over $D_{KL}$ will not significantly improve reconstruction but will lead to a waste of bitrate (tokens). We provide an additional in Table 18, showing that GQ trained with 14 bits and quantized into 18 bits does not perform as well as GQ trained with 18 bits and quantized into 18 bits. The drawback of simply increasing the codebook size for GQ is that it is not as effective as increasing the target of TDC to that size and quantizing with a proper $\log K = D_{KL}$.

Table 18: The effect of simply increase codebook size.

| Training Target | Codebook Size | bpp (num of tokens) | PSNR↑ | LPIPS↓ | SSIM↑ | rFID↓ |
|---|---|---|---|---|---|---|
| $D_{KL}(2) = 14$ | $\log_2 K = 14$ | $2^{14} \times 1024$ | 25.31 | 0.064 | 0.762 | 0.491 |
| $D_{KL}(2) = 14$ | $\log_2 K = 18$ | $2^{18} \times 1024$ | 27.79 | 0.059 | 0.808 | 0.513 |
| $D_{KL}(2) = 18$ | $\log_2 K = 18$ | $2^{18} \times 1024$ | 27.86 | 0.054 | 0.804 | 0.424 |

## D.7 QUANTIZED LATENT VISUALIZATION

In Figure 4, we show the t-NSE (van der Maaten & Hinton, 2008) visualization of latent after GQ, using 5 subclass of ImageNet.

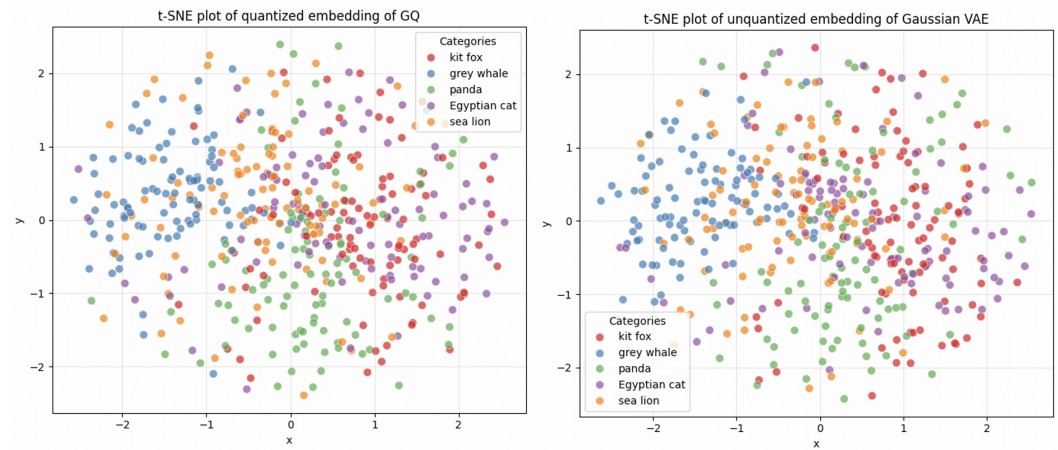

Figure 4: The t-NSE visualization of latent of GQ vs. unquantized Gaussian VAE. It is shown that the latent before and after quantization are quite similar.

Table 19: The generation performance of GQ in different bitrate regime.

| Method | bpp (num of tokens) | gFID | IS |
|---|---|---|---|
| TokenBridge | $0.1875(2^{16} \times 256)$ | 8.29 | 188.05 |
| GQ (Ours) | $0.1875(2^{16} \times 256)$ | 7.74 | 229.53 |
| TokenBridge | $0.25(2^{16} \times 256)$ | 7.82 | 198.24 |
| GQ (Ours) | $0.25(2^{16} \times 256)$ | 7.67 | 230.79 |

### D.8 MORE GENERATION RESULTS

To better understand the generation performance, in Table 19, we present the auto-regressive generation result of GQ in different bitrate regime. And in Table 20, we present the auto-regressive generation result of GQ in FFHQ dataset. It is shown that the advantage of GQ is consistent in different bitrate regime and datasets.

### D.9 PRIOR-POSTERIOR MISMATCH

Sometimes the Gaussian VAE might suffer from prior-posterior mismatch. However, in our case, such mismatch is not severe. To illustrate this, we estimate the prior posterior mismatch by considering the relationship between $q(Z)$ and $\mathcal{N}(0, I)$. More specifically, we have

$$D_{KL}(q(Z)||N(0,1)) \approx \frac{1}{N} \sum_{i=1}^{N} (\log q(z^i) - \log N(z^i|0, I)). \tag{33}$$

Additionally, we can estimate the optimal bitrate without effected by prior posterior mismatch with a similar approximation:

$$D_{KL}(q(Z|X)||p(Z)) \approx \frac{1}{N} \sum_{i=1}^{N} (\log q(z^i|X) - \log q(z^i)). \tag{34}$$

Table 20: The generation performance of GQ with FFHQ dataset.

| Method | bpp (num of tokens) | gFID | IS |
|---|---|---|---|
| BSQ | $0.25(2^{16} \times 256)$ | 5.48 | - |
| TokenBridge | $0.25(2^{16} \times 256)$ | 7.15 | - |
| GQ (Ours) | $0.25(2^{16} \times 256)$ | 5.09 | - |

Table 21: The bitrate and prior-posterior mismatch.

| Divergence | bits per pixel |
|---|---|
| bpp *w.r.t.* $\mathcal{N}(0, I)$ ($D_{KL}(q(Z\|X)\|\|\mathcal{N}(0,1))$) | 0.25 |
| bpp *w.r.t.* $q(Z)$ ($D_{KL}(q(Z\|X)\|\|q(Z))$) | 0.25 |
| prior posterior mismatch ($D_{KL}(q(Z)\|\|N(0,1))$) | 0.000328 |

We train a diffusion model to estimate $\log q(z^i)$ by using PF-ODE and Skilling-Hutchinson trace estimator (See Appendix D.2 of Song et al. (2020)). We use the Gaussian VAE (w/ TDC) + DiT diffusion model and ImageNet validation dataset. The euler PF-ODE steps is set to 250 and the Skilling-Hutchinson number of sample is set to 1. The result is shown in Table 21. The results show that the prior-posterior mismatch is only 0.00033 bits per pixel, accounting for approximately 0.1% of the total bpp. Furthermore, the best bpp and the actual bpp show no significant difference on a scale of 0.01. This indicates that the "bitrate waste" caused by the prior-posterior mismatch is negligible, and the mismatch itself is not significant.

## D.10 COMPLEXITY

Table 22: The encoding and decoding overhead of GQ over Gaussian VAE.

| Method | UNet based | |
|---|---|---|
| | Encoding FPS | Decoding FPS |
| Gaussian VAE | 104 | 64 |
| GQ (torch) | 12 | 61 |
| GQ (CUDA) | 79 | 61 |

As with FSQ and BSQ (Mentzer et al., 2023; Zhao et al., 2024), our codebook can be generated on the fly by maintaining the same random number generator seed on both the encoder and decoder sides. Therefore, our GQ model has the same parameter size as the vanilla Gaussian VAE. In Table 22, we compare the encoding and decoding frames per second (FPS) of the Gaussian VAE and GQ. We use $256 \times 256$ images with a batch size of 1, and we report the wall clock time, meaning that the time required for loading data is included. The results show that the encoding FPS of GQ (implemented in PyTorch) is 12 on an H100 GPU, which is considerably slower than the 104 FPS achieved by the Gaussian VAE. On the other hand, GQ does not incur any decoding overhead.

To reduce the computational complexity of GQ, we implement GQ using a tailored CUDA kernel. Specifically, we follow the approach of Vonderfecht & Liu (2025), with a key difference: we maintain the codebook, as our bottleneck is not codebook instantiation. Additionally, we avoid the creation of large buffer vectors by performing the summation over $m$ within the CUDA kernel instead of in PyTorch. With this approach, we achieve an encoding FPS of approximately 80, with negligible overhead compared to the Gaussian VAE. A detailed comparison between the PyTorch implementation and the CUDA implementation of GQ is provided below as `GQ_torch` and `GQ_CUDA`, respectively.

```python
def GQ_torch(mu, sigma, codebook, m, bs, K):
    # mu.shape = (bs, m)
    # sigma.shape = (bs, m)
    # codebook.shape = (K, m)

    # This step create (bs, m, K) vector, which is the performance
    bottleneck
    dist_m =((mu[:,None] - codebook[None])/sigma[:,None])**2
    dist = torch.sum(dist_m, dim=1) # sum over m dimension
    indices = torch.argmin(dist, dim=1) # argmax over K dimension
    zhat = torch.index_select(codebook, 0, indices) # select quantized
    results
    return indices, zhat
```

```
1  def GQ_CUDA(mu, sigma, codebook, m, bs, K):
2      dist = torch.zeros([bs, K])
3      # need an extension wrapping and register the kernel into operator,
       we omit it in paper
4      # see code appendix for details
5      GQ_Kernel<<<bs * K / 256,256>>>(mu, sigma, codebook, dist, m, bs, K)
6
7      indices = torch.argmin(dist, dim=1) # argmax over K dimension
8      zhat = torch.index_select(codebook, 0, indices) # select quantized
       results
9      return indices, zhat
10
11 __global__ void GQ_Kernel(
12   const float* mu,
13   const float* sigma,
14   const float* codebook,
15   float* dist,
16   int64_t m,
17   int64_t bs,
18   int64_t K
19 ) {
20   int idx = blockIdx.x * blockDim.x + threadIdx.x;
21   if (idx >= K * bs) return;
22   int bi = idx / K;
23   int ni = idx % K;
24   float a = 0.0f;
25   for (int i = 0; i < m; i++) {
26       float b = (codebook[ni * m + i] - mu[bi * dim + i]) / sigma[bi *
       dim + i];
27       a += b * b;
28   }
29   dist[idx] = a;
30   return;
31 }
```

### D.11 ASYMPTOTIC COMPLEXITY

It is noteworthy that GQ without grouping quantization, such as PT or TR, or GQ with a group size of $m = 1$, is asymptotically faster than reverse channel coding methods. This is because, for $m = 1$, the GQ target in Eq.3 reduces to a quadratic form. In this case, it suffices to sort the scalar codebook $c_{1:K}$ in advance. Despite the sorting takes $\Omega(D_{KL}(q(Z_i|X)||\mathcal{N}(0,1)))$, the sorting is only need to be done once and can be amortized across dimension and dataset. Subsequently, the minimization in Eq.3 can be performed in $O(D_{\mathrm{KL}}(q(Z_i|X)||\mathcal{N}(0,1)))$ time using binary search. The details is shown in Algorithm 2.

On the other hand, most reverse channel coding methods require $O(2^{D_{\mathrm{KL}}(q(Z_i|X)||\mathcal{N}(0,1))})$ computational complexity (Havasi et al., 2018b; Flamich et al., 2020; Theis & Yosri, 2021). A$^*$ coding (Flamich et al., 2022) can achieve $O(D_\infty(q(Z_i|X)||\mathcal{N}(0,1)))$ encoding complexity, albeit at the cost of increased decoding complexity.

However, we note that this complexity advantage is not particularly meaningful in practice. This is because any auto-regressive generation model requires a softmax operation over the entire codebook, which has a complexity of $O(2^{D_{\mathrm{KL}}(q(Z_i|X)||\mathcal{N}(0,1))})$. In practice, only tractable codebook sizes, such as $2^{16}$ or $2^{18}$, are used.

# E  ADDITIONAL QUANTITATIVE RESULTS

## E.1  ADDITIONAL QUALITATIVE RESULTS AND FAILURE CASES

In Figure 5, we present additional qualitative results showing that GQ achieves superior visual quality. However, we also note that none of the approaches is successful in reconstructing the plate of the residential vehicle. The text content remains challenging for low bitrate VQ-VAEs.

---

**Algorithm 1:** GQ (argmax)

1 **input** $c_{1:K}$ (sorted, such that $c_j \leq c_{j+1}$), $\mu_i$
2    $\mathcal{T}^* = \infty$
3    **for** $j = 1$ **to** $K$ **do**
4      **if** $\mathcal{T}^* \leq ||c_j - \mu_i||$ **do**
5        $\mathcal{T}^* = ||c_j - \mu_i||, j^* = j$
6    **return** $c_{j^*}, j^*$

**Algorithm 2:** GQ (bisect)

1 **input** $c_{1:K}$ (sorted, such that $c_j \leq c_{j+1}$), $\mu_i$
2    $l = 1, r = K$
3    **while** $l + 1 < r$ **do**
4      $m = (l + r)//2$
5      **if** $c_m < \mu_i$ **do**
6        $l = m$
7      **else**
8        $r = m$
9    **if** $||c_l - \mu_i|| < ||c_r - \mu_i||$ **do**
10      **return** $c_l, l$
11    **else**
12      **return** $c_r, r$

---

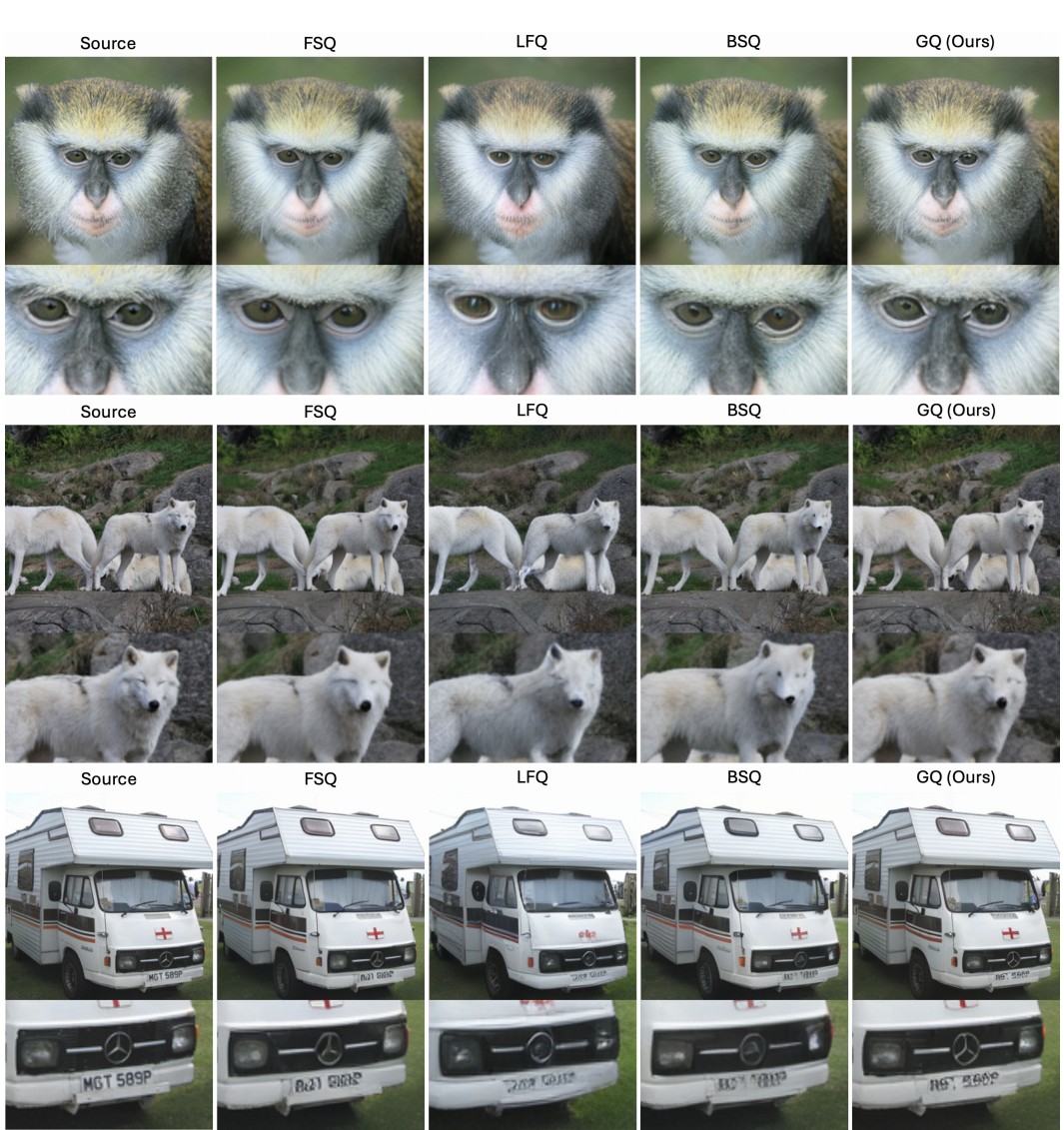

Figure 5: Qualitative results on ImageNet dataset and 0.25 bpp. None of those approaches correctly reconstruct the plate number.

