# OpenReview forum: "Vector Quantization using Gaussian Variational Autoencoder"
_ICLR.cc/2026/Conference — Submitted to ICLR 2026_

### Official Review · Reviewer_MSjv · 2025-10-18

**Soundness:** 3
**Presentation:** 1
**Contribution:** 2
**Rating:** 2
**Confidence:** 3

**Summary:**

This paper proposes a latent discretization method that converts a (continuous latent) Gaussian VAE into a VQ-VAE without requiring additional codebook training. The authors introduce a simple codebook generation and code selection mechanism, supported by theoretical analysis regarding the appropriate codebook size. Additionally, they propose an adaptive KL regularization for VAE training, which encourages the Gaussian VAE to have a latent capacity closer to a pre-defined bits-back coding bitrate.

**Strengths:**

- The theorems provide valuable insights for estimating the appropriate codebook size.
- The introduction of the Target Divergence Constraint is novel and interesting.
- The reconstruction performance improvement of GQ over existing methods is impressive.

**Weaknesses:**

**Motivation for conversion is not clearly presented**
- The motivation for the proposed conversion is not sufficiently clear. Since the training framework begins with a pre-trained Gaussian VAE, the benefits of using the resulting VQ-VAE over the original Gaussian VAE should be discussed in greater detail.

**Some aspects of the methodology are unclear**
- It is not clear whether the codebook depends on $i$ throughout the manuscript.
- The timing of codebook $c_{1:K}$ sampling is ambiguous. Providing pseudo-code or an algorithmic description would help clarify this point.
- Section 3.4 does not clearly explain the necessity of grouping. Even after multiple readings, the rationale remains unclear. If $\log_2 K = 0$ for some $i$, wouldn’t preparing a single code for that dimension suffice?

**Comparison with TokenBridge**
- TokenBridge should be described in more detail, as it is the most relevant related work.
- The proposed method should be compared with TokenBridge more thoroughly, not just in terms of generation performance. Section 3.5 is not accessible to readers unfamiliar with TokenBridge.
- What are the key factors that contribute to the improved performance of the proposed method compared to TokenBridge?

**Experimental verification**
- The experiments are somewhat limited regarding generation performance, which is only reported in Table 7. In particular, a direct comparison of the VQ-VAE learned by the proposed method with the original Gaussian VAE in terms of generation performance would be valuable.
- The performance of LFQ is significantly worse than the other baselines. It is unclear whether the experiments are fair. Were the main experiments conducted using existing codebases, or did you implement the code yourselves?

**Minor weaknesses**
- Equation (1) is not a standard formulation, as it does not include commitment and codebook losses.
- $\sigma_j$ in Theorem 1 should be $\sigma_i$?
- The reference for TokenBridge is incorrect in line 219.

**Questions:**

- From Theorems 1 and 2, I understand that $\log K$ is expected to be larger than the KL divergence, and that a larger $\log K$ is beneficial for improving generation performance. However, line 154 states that $\log K$ should be close to the KL divergence. Is there any drawback to simply increasing the codebook size?
- According to the manuscript, the codebook is generated by random sampling from a Gaussian distribution, which seems to make the results highly dependent on the random seed. Is this correct?
- While Gaussian VAEs typically suffer from posterior–prior mismatch, the codebook is sampled from the prior (normal) distribution. Do you have any insights on whether sampling the codebook from this distribution is sufficiently effective?

**Details Of Ethics Concerns:**

Potential ethic concerns are discussed in a section.

---

> ### Author Response · Authors · 2025-11-13
> **Response to the comments of Reviewer MSjv Part I**
>
> Thank you for your detailed review. We are pleased to provide answers to your questions:
>
> * __W1. ...the benefits of using the resulting VQ-VAE over the original Gaussian VAE should be discussed in greater detail...__: The benefits of using VQ-VAE over the original Gaussian VAE are very clear: only VQ-VAE produces discrete tokens and can be used for autoregressive generation (AR or MIM).
>
> * __W2. ...not clear whether the codebook depends on i...__: VQ-VAE requires the same codebook for all dimensions. Therefore, our codebook does not depend on i. For all $i\in[1,d]$, our codebook has the same size $K$ and content $c_{1:K}$.
>
> * __W3. The timing of codebook sampling is ambiguous. Providing pseudo-code or an algorithmic description would help clarify this point...__: The pseudocode and algorithm of GQ are already provided in Appendix D.4 and Appendix D.5. The codebook sampling, if you mean codebook construction, is simply sampling iid samples from standard Gaussian distribution. For $K$ size codebook with each vector $m$ dimension, the iid Gaussian sample takes $O(mK)$ to draw. The codebook is constructed only once and then fixed. It is implemented in this way:
>   ```python
>     import torch
>     from torch.quasirandom import SobolEngine
>     from scipy.stats import norm
>
>     sobol = SobolEngine(m, scramble=True, seed=seed)
>     samples_sobol = sobol.draw(K)
>     codebook = torch.from_numpy(norm.ppf(samples_sobol))
>   ```
>
> * __W4. Section 3.4 does not clearly explain the necessity of grouping...__: The original VQ-VAE has three key parameters: the number of tokens, the codebook size, and the codebook dimension. As we construct our VQ-VAE from a Gaussian VAE, we want our converted model to have exactly the same controllable parameters as the original VQ-VAE. Without grouping, we can only control the number of tokens and the codebook size, and the codebook dimension is always 1. With grouping, we can additionally control the codebook dimension, as the codebook dimension equals the group size. Besides, grouping also enhances the reconstruction results. In Table 12, we show that grouping with m=4 (grouping four tokens into a large one) improves reconstruction over grouping with m=1 (no grouping). The log K = 0 case is an extreme example to demonstrate the benefits of grouping, as in this scenario, all reconstructions would be identical and the reconstruction performance would be very poor.
>
> * __W5. TokenBridge should be described in more detail...__: We will expand the discussion of TokenBridge in related works: "... TokenBridge and ReVQ also convert a pre-trained Gaussian VAE into a VQ-VAE. TokenBridge adopts the Post Training Quantization (PTQ) technique from model compression and proposes to treat latent $z$ as model parameters to discretize. It uses a fixed codebook composed of K centroids of a Gaussian distribution, which is similar to the historgram used in PTQ. It then quantizes the posterior sample by finding the closest centroid. However, TokenBridge directly quantizes a vanilla Gaussian VAE without limiting the KL divergence of each dimension to be the same. This leads to suboptimal rate-distortion performance as $K$ is likely to be too large for some dimension and too small for some dimension considering the $D_{KL}$ variation across different dimensions. Thus, TokenBridge can be improved by TDC, which enforces that constraint. ...."
>
> * __W6. The proposed method should be compared with TokenBridge more thoroughly, not just in terms of generation performance...__: We have compared GQ with TokenBridge comprehensively in terms of reconstruction performance in Table 3 and Table 11. These results include standard metrics such as PSNR, LPIPS, SSIM, and rFID; they are not limited to generation performance. If you mean "not just in terms of reconstruction performance" instead of "not just in terms of generation performance", we also provide a generation performance of TokenBridge. We adopt the unconstrained Gaussian VAE as Table 3, and the same AR generative model in Table 4. The result shows that using the same AR model and token size, TokenBridge is outperfomred by GQ.
>
>     |             | bpp (num of token)        | gFID | IS |
>     |-------------|---------------------------|------|----|
>     | TokenBridge | $0.25 (2^{16}\times 256)$ | 7.82 | 198.24 |
>     | GQ          | $0.25 (2^{16}\times 256)$ | 7.67 | 230.79 |

---

> ### Author Response · Authors · 2025-11-13
> **Response to the comments of Reviewer MSjv Part II**
>
> * __W7. ...the key factors that contribute to the improved performance of the proposed method compared to TokenBridge...__: As explained in Sec. 3.5 and shown in Table 3, TokenBridge directly quantizes a vanilla Gaussian VAE, whose KL divergence varies significantly. TokenBridge uses the same K across all dimensions, and since the KL divergence varies violently across dimensions, it is inevitable that for some dimensions, K is much larger than the KL divergence, while for others, K is much smaller. When K is much larger than the divergence, the bitrate is wasted. When K is much smaller than the divergence, a large quantization error emerges. Our TDC improves upon TokenBridge by providing a Gaussian VAE with the same KL divergence across each dimension.
>
> * __W8. ...a direct comparison of the VQ-VAE learned by the proposed method with the original Gaussian VAE in terms of generation performance would be valuable...__: In Table 4, the generation performance of all the methods (VQGAN, LFQ, BSQ, GQ) are compared using the same auto-regressive generative model with the same architecture. The Gaussian VAE cannot be used for auto-regressive generation, so such comparison is impossible. If you mean "reconstruction results" such as PSNR, LPIPS, DISTS and rFID, it is already shown in Table 6-7.
>
> * __W9. ...The performance of LFQ is significantly worse than the other baselines. It is unclear whether the experiments are fair. Were the main experiments conducted using existing codebases...__: The LFQ paper is closed-source [Language Model Beats Diffusion -- Tokenizer is Key to Visual Generation]. Furthermore, the original LFQ paper does not provide results for image reconstruction. Therefore, we adopt the LFQ implementation from the BSQ paper, using the BSQ codebase [Image and Video Tokenization with Binary Spherical Quantization]. In fact, our results are aligned with the LFQ results reported in the BSQ paper. The reconstruction performance of LFQ in Table 5 of the BSQ paper is significantly worse than the other baselines. To compensate for this, we additionally show results of GQ compared to OpenMagViT-V2 [Open-MAGVIT2: An Open-Source Project Toward Democratizing Auto-regressive Visual Generation
> ], which is the best open-source implementation of LFQ. Our GQ achieves better PSNR, SSIM, and rFID than OpenMagViT-V2 (LFQ), with much less training time.
>
>     |                           | bpp (num of tokens) | PSNR  | LPIPS | SSIM  | rFID | Training Epoches |
>     |---------------------------|---------------------|-------|-------|-------|------|------------------|
>     | OpenMagViT-V2 (LFQ) | $2^{18}\times 256$    | 21.63 | 0.111 | 0.640 | 1.17 | 350              |
>     | GQ (Ours)                 | $2^{18}\times 256$    | 22.30 | 0.116 | 0.642 | 1.04 | 40               |
>
> * __W10. ...Equation (1) is not a standard formulation, as it does not include commitment and codebook losses...__:
> Eq. 1 is the abstract rate distortion target of VQ-VAE, we will extend Eq. 1 into:
>     $$\log K + E[\Delta(X,g(\hat{z}))] + \mathcal{L}\_{reg},$$
>     and explain that the $\mathcal{L}_{reg}$ might be composed of the commitment loss and the codebook loss from [Neural discrete representation learning], or the Gumbel softmax loss from [Continuous Relaxation Training of Discrete Latent Variable Image Models].
>
> * __W11. $\sigma_j$ in Theorem 1 should be $\sigma_i$__: Yes indeed, thanks for pointing that out and we will fix this typo.
>
> * __W12. The reference for TokenBridge is incorrect in line 219__: Thanks for pointing that out, we will fix the reference to [Bridging continuous and discrete tokens for autoregressive visual generation].
>
> * __Q1. Is there any drawback to simply increasing the codebook size?__: According to Theorem 1, increasing $\log K$ too much over $D_{KL}$ will not significantly improve reconstruction but will lead to a waste of bitrate (tokens). We provide an additional result below, showing that GQ trained with $14$ bits and quantized into $18$ bits does not perform as well as GQ trained with $18$ bits and quantized into $18$ bits. The drawback of simply increasing the codebook size for GQ is that it is not as effective as increasing the target of TDC to that size and quantizing with a proper $\log K=D_{KL}$.
>
>     |                                | bpp (num of tokens) | PSNR | LPIPS | SSIM |rFID|
>     |--------------------------------|---------------------|---|---|---|---|
>     | GQ Trained with $D_{KL}=14$ Quant into $\log K=14$ | $2^{14}\times 1024$ | 25.31 | 0.064 | 0.762 | 0.491 |
>     | GQ Trained with $D_{KL}=14$ Quant into $\log K=18$ | $2^{18}\times 1024$ | 27.79 | 0.059 | 0.808 | 0.513 |
>     | GQ Trained with $D_{KL}=18$ Quant into $\log K=18$ | $2^{18}\times 1024$ | 27.86 | 0.054 | 0.804 | 0.424 |

---

> ### Author Response · Authors · 2025-11-13
> **Response to the comments of Reviewer MSjv Part III**
>
> * __Q2. the codebook is generated by random sampling from a Gaussian distribution, which seems to make the results highly dependent on the random seed__: This is incorrect, as the codebook is usually large enough ($2^{16}$) to compensate for the randomness. We provide additional results below showing that the random seed has little effect on reconstruction performance. We used three continous random seeds (42,43,44) without cherry-picking. It is clear that the performance of GQ is not affected by randomness.
>
>     |            | bpp (num of tokens) | PSNR | LPIPS | SSIM | rFID |
>     |------------|------|------|-------|------|------|
>     | GQ Seed=42 | $2^{16}\times 1024$ | 27.61 | 0.059 | 0.807 | 0.529 |
>     | GQ Seed=43 | $2^{16}\times 1024$ | 27.61 | 0.059 | 0.807 | 0.523 |
>     | GQ Seed=44 | $2^{16}\times 1024$ | 27.62 | 0.059 | 0.807 | 0.526 |
>
> * __Q3: While Gaussian VAEs typically suffer from posterior–prior mismatch, the codebook is sampled from the prior (normal) distribution__:
>   * For quantization and compression, if in some region the $N(0,1)$ distribution has mass but $q(z)=\int p(x)q(z|x)dx$ does not, this is acceptable as it only wastes some bitrate. However, if in some region $q(z)$ has mass but $N(0,1)$ does not, this is very bad as this means large quantization error. For sampling from a Gaussian VAE, if in some region $N(0,1)$ has mass but $q(z)$ does not, this is bad because a poor sample is generated. However, if in some region $q(z)$ has mass but $N(0,1)$ does not, this is acceptable as it means we are only dropping a mode.
>
>   * The Gaussian VAE's objective is $D_{KL}(q(z|x)||N(0,1))$, which penalizes $q(z|x)$ for having probability mass where $N(0,1)$ does not, but not the reverse. Therefore, the posterior-prior mismatch is not a severe problem for quantization and compression, but it is a severe problem for directly sampling from a Gaussian VAE.
>
>   * On the other hand, the posterior-prior mismatch is not severe for our GQ Gaussian VAE. As evidence, we conducted a normality test on our latent variables over the first 1000 samples of the ImageNet validation split, using the GQ Gaussian VAE with $2^{16}\times 1024$ tokens. The result of the Shapiro-Wilk test shows a statistic of 0.99 and a p-value of 0.74, which indicates that our sample is Gaussian.

---

> > ### Comment · Reviewer_MSjv · 2025-11-23
> > **Post-rebuttal comments (1/2)**
> >
> > Thank you for your prompt rebuttal. While some of my questions have been partially addressed, several concerns remain from my perspective. In general, some of your responses are too brief, which makes it difficult for me to fully understand the main points of some of each reply or to be convinced by your arguments. Please find my post-rebuttal comments below.
> >
> > ---
> > ### The current manuscript still lacks convincing motivation
> >
> > > The benefits of using VQ-VAE over the original Gaussian VAE are very clear...
> >
> > Your brief response did not fundamentally address my concern. I understand that you stated that VQ-VAE can produce discrete tokens that can be learned by autoregressive models, whereas continuous-latent VAEs cannot utilize this family of generative models. While this is accurate, the use of autoregressive models alone does not constitute a strong motivation, particularly in the image domain, where there are numerous high-performing generative models based on continuous latents. For example, the BridgeToken paper provides a convincing motivation for quantization techniques by proposing BridgeToken to bridge the gap between VQ-VAE and VAE.
> >
> > > The Gaussian VAE cannot be used for auto-regressive generation, so such comparison is impossible.
> >
> > In fact, the TokenBridge paper includes such a comparison: the authors compare their method with four VAE + latent diffusion models of comparable parameter size. This comparison is crucial for their motivation, as it bridges the gap between VQ-VAE and Gaussian VAE. Again, converting a Gaussian VAE to a VQ-VAE for improved autoregressive generation is a weak motivation. If the proposed method underperforms the Gaussian VAE, such quantization is not meaningful; in that case, one could simply use the original Gaussian VAE. Additionally, since the strength of the TDC constraint directly affects the latent distribution, a comparison between your method and Gaussian VAE + latent diffusion under different TDC strengths would be insightful.
> >
> > ---
> > ### Experiments regarding generation performance
> >
> > > We also provide generation performance of TokenBridge.
> >
> > First of all, I apologize for the typo in my initial review. My intention was to inquire about generation performance.
> >
> > As **Reviewer uBPL** also pointed out, the generation experiments are limited, whereas reconstruction performance is extensively evaluated. You stated in response to the reviewer’s comment that your current focus is on vector quantization rather than image generation. However, evaluation in terms of generation performance is crucial in the image domain, since image generation is the primary application of vector quantization, unlike in audio applications such as codecs.
> >
> > I agree with your statement that a tokenizer’s generation performance is not strongly correlated with generation quality. I believe this characteristic further highlights the importance of testing vector quantization methods in generation tasks. From my perspective, the generation experiments in the current manuscript are limited in the following respects:
> > - As **Reviewer uBPL** noted, more diverse datasets should be evaluated.
> > - The proposed method (and BridgeToken) should be applied to generation tasks with different bpp settings.
> >
> > ---
> > ### Prior–posterior mismatch
> >
> > > If in some region the $N(0,1)$ distribution has mass but ..., this is acceptable as it only wastes some bitrate.
> >
> > I believe that bitrate wastage is not a trivial issue in VQ-VAE-related research. Some analysis quantifying the impact of mismatch on performance should be provided if bitrate is indeed wasted. Otherwise, you should clarify why bitrate wastage is not problematic, or demonstrate that bitrate wastage does not occur in practice. The following analyses could be considered:
> > - Compare the codebook obtained by quantizing $N(0,1)$ and $q(z)$, for example, as approximated by a pre-trained latent diffusion model.
> > - Report additional metrics such as precision and recall in generation experiments.
> > - Apply statistical tools that measure distributional differences in high-dimensional spaces.
> >
> > > As evidence, we conducted a normality test on our latent variables over the first 1000 samples of the ImageNet validation split....
> >
> > The experimental setup is not entirely clear for this purpose. Did you use discrete points obtained by quantizing ImageNet latents by $2^{16}$? If your goal is to check whether the ImageNet latent distribution follows a Gaussian, such quantization should not be involved. Furthermore, significant prior–posterior mismatch is typically observed in standard Gaussian VAEs; otherwise, second-stage training (e.g., diffusion on VAE latents) would not be necessary. Additionally, your response does not clarify how the Shapiro-Wilk statistic was calculated. Was it computed for each dimension independently, ignoring inter-dimensional correlations, or was the test applied to the high-dimensional latent vectors as a whole?

---

> ### Comment · Reviewer_MSjv · 2025-11-23
> **Post-rebuttal comments (2/2)**
>
> ### Additional comments in response to Reviewer uBPL's comment
>
> I agree with their first concern regarding the prerequisite of "training a Gaussian VAE with TDC." While you have minimally revised the corresponding section, you should more clearly state that your proposed method generally requires VAE training from scratch, as this is a necessary stage to benefit from the training-free conversion.

---

> ### Author Response · Authors · 2025-11-27
> **Response to the comments of Reviewer MSjv Part IV**
>
> * __1 The current manuscript still lacks convincing motivation__:
> * Thank you for the further clarification. We also provide generation results from a Gaussian VAE + DiT model, both with and without TDC. We adopt the standard DiT on ImageNet, with the model and training settings matching our previous AR model (ImageNet 128x128, 100 epochs, 250-step Euler ODE, DiT layers=16, hidden_size=1024, num_heads=16). The results show that when computation is aligned, the AR model is more efficient than the diffusion model. This finding strengthens the motivation for using VQ-VAE over a Gaussian VAE. Furthermore, TDC does not significantly change the Gaussian VAE's generation performance. However, we also note that discrete latent space is valuable in itself, as it serves as a representation and codec, regardless of generation performance.
>
>     |             | bpp (num of token)        | gFID | IS |
>     |-------------|---------------------------|------|----|
>     | Gaussian VAE (w/o TDC) + DiT | $0.25 (-)$ | 8.35 | 202.19 |
>     | Gaussian VAE (w/ TDC) + DiT | $0.25 (-)$ | 8.47 | 205.94 |
>     | TokenBridge | $0.25 (2^{16}\times 256)$ | 7.82 | 198.24 |
>     | GQ          | $0.25 (2^{16}\times 256)$ | 7.67 | 230.79 |
>
> * __2 Experiments regarding generation performance__:
> * Thank you for the advice regarding different bits-per-pixel (bpp) rates. To better understand the generation performance of GQ and TokenBridge at different bpp levels, we have provided additional results. They show that a lower bpp ($2^{12}$) is outperformed by a higher bpp ($2^{16}$), and that GQ consistently outperforms TokenBridge.
>
>     |             | bpp (num of token)        | gFID | IS |
>     |-------------|---------------------------|------|----|
>     | TokenBridge | $0.1875 (2^{12}\times 256)$ | 8.29 | 188.05 |
>     | GQ | $0.1875 (2^{12}\times 256)$ | 7.74 | 229.53 |
>     | TokenBridge | $0.25 (2^{16}\times 256)$ | 7.82 | 198.24 |
>     | GQ | $0.25 (2^{16}\times 256)$ | 7.67 | 230.79 |
>
> * For different datasets, we test the performance of BSQ, GQ and TokenBridge on FFHQ dataset with the same setup as ImageNet. The result is consistent with ImageNet dataset and shown below:
>
>     |             | bpp (num of token)        | gFID | IS |
>     |-------------|---------------------------|------|----|
>     | TokenBridge | $0.25 (2^{16}\times 256)$ | 5.48 | - |
>     | BSQ | $0.25 (2^{16}\times 256)$ | 7.15 | - |
>     | GQ | $0.25 (2^{16}\times 256)$ | 5.09 | - |
>
> * __3 Prior–posterior mismatch__:
> * As suggested by the reviewer, to numerically show the prior-posterior mismatch, we estimate the divergence between q(Z) and N(0,1) by training a latent diffusion model. More specifically, we have:
> $$
> D_{KL}(q(Z)||N(0,1)) \approx \frac{1}{N}\sum_{i=1}^N (\log q(z^i) - \log N(z^i|0,I)).
> $$
> * Additionally, we can estimate the optimal bitrate without effected by prior posterior mismatch with a similar approximation:
> :
> $$
> D_{KL}(q(Z|X)||p(Z)) \approx \frac{1}{N}\sum_{i=1}^N (\log q(z^i|X) - \log q(z^i)).
> $$
> * We train a diffusion model to estimate $\log q(z^i)$ by using PF-ODE and Skilling-Hutchinson trace estimator (See Appendix D.2 of [Score-Based Generative Modeling through Stochastic Differential Equations
> ]) following the implementation of SiT (https://github.com/willisma/SiT/blob/main/transport/transport.py). We use the "Gaussian VAE (w/ TDC) + DiT" diffusion model in our first response. The prior posterior mismatch on ImageNet validation dataset. The euler PF-ODE steps is set to 250 and the Skilling-Hutchinson number of sample is set to 1. The result is shown as follows:
>
>     | Name                                               | Bitrate (bits / pixel) |
>     |----------------------------------------------------|------------------------|
>     | bpp wrt $N(0,I)$ ($D_{KL}(q(Z\|X)\|\|N(0,1))$) | 0.25 |
>     | bpp wrt $q(Z)$ ($D_{KL}(q(Z\|X)\|\|q(Z))$) | 0.25 |
>     | prior posterior mismatch ($D_{KL}(q(Z)\|\|N(0,1))$)  | 0.000328 |
>
> * The results show that the prior-posterior mismatch is only 0.00033 bits per pixel, accounting for approximately 0.1% of the total bpp. Furthermore, the best bpp and the actual bpp show no significant difference on a scale of 0.01. This indicates that the "bitrate waste" caused by the prior-posterior mismatch is negligible, and the mismatch itself is not significant.
>
> * No quantization is involved in this analysis. The previous Shapiro-Wilk test was conducted on an unquantized Gaussian VAE with a TDC constraint, with the statistical test computed by treating each dimension independently.
>
> * __4 Additional comments in response to Reviewer uBPL's comment__: Thank you for your suggestion. We will state more clearly that GaussianQuant requires training a Gaussian VAE with a specific constraint. In the Introduction and Method section, we will emphasize that a good reconstruction performance stems from a Gaussian VAE trained with TDC.

---

### Official Review · Reviewer_ym9y · 2025-10-30

**Soundness:** 3
**Presentation:** 3
**Contribution:** 3
**Rating:** 6
**Confidence:** 3

**Summary:**

This paper proposes a framework for obtaining a VQ-VAE from a Gaussian VAE.
The proposed Gaussian Quantization (GQ) method generates a random Gaussian noise codebook and utilizes it to quantize the posterior mean of the Gaussian VAE encoder.
The authors present theoretical analyses that clarify the relationship between the bits-back coding rate of Gaussian VAEs and the required codebook size for achieving small quantization error.
Based on these analyses, they also propose the Target Divergence Constraint (TDC) to train Gaussian VAEs suitable for GQ, as well as grouping techniques that facilitate GQ by organizing the dimensions of the latent variables.
Experimental results demonstrate that the proposed framework outperforms baseline methods.

**Strengths:**

1. The paper is well structured and is easy to follow.
2. The idea of discussing the relationship between quantization error and bits-back encoding is interesting.
3. The effectiveness of the proposed method is consistently demonstrated in the experiments.

**Weaknesses:**

1. The method for determining the $\lambda$ parameters in TDC seems questionable. According to Eq. (6) and (7), when there is no dimension to which $\lambda_\mathrm{min}$ and $\lambda_\mathrm{max}$ are applied, $\lambda_\mathrm{min}$ and $\lambda_\mathrm{max}$ are multiplied by 0.99 at each step. Is there a risk that these parameters become extremely small when needed again?
2. Further explanation is needed for why Eq. (8) encourages greater codebook usage and entropy, and why Eq. (8) takes its specific form. For example, if the first term were a squared L2 norm, it would resemble the form of $\log p(c_j|x)$ for a Gaussian, but this is not the case. While some details seem to be described in Appendix C.2, a clearer explanation in the main text would be helpful.
3. The connection between IsoKL, MIRACLE, and the proposed method at line 246 is not sufficiently clear. Although there is some explanation around line 401, a more explicit discussion and guidance earlier in the text would improve readability.
4. Although Theorems 1 and 2 provide conditions for large and small quantization errors, the quantization error is measured in latent space and distortion caused by quantization in data space is not discussed. When the Lipschitz constant of decoder is large, one concern is that small quantization errors can be magnified to large distortions in data space.

Other minor comments:
* The definition of distortion in Eq. (1) is unclear, especially regarding the commitment loss [Van Den Oord et al., 2017]. If the commitment loss is intentionally excluded, it should be explicitly stated. If the commitment loss is included in the distortion term, then $\Delta$ should be expressed as a function of $z$ as well as $X$ and $g(z)$.
* In the conditions of Theorem 1, $\mu_i \sigma_j \leq c_1$ and $|\mu_i| + |\sigma_j| \leq c_2$, both $i$ and $j$ are used. Isn't this a typo? Please clarify whether this is a typo or if different variables are intended.

[Van Den Oord et al., 2017] Aaron Van Den Oord, Oriol Vinyals, et al. Neural discrete representation learning. Advances in neural information processing systems, 30, 2017.

**Questions:**

1. Could you give an answer to the question in Weakness 1? Or is there any ablation study regarding this algorithm design choice of the adaptive weights $\lambda_i ~ (i=1,\dots,d)$?
2. Could you give further explanations regarding Weaknesses 2 and 3?
3. Regarding Weakenss 4: Have you considered evaluating the distortion caused by quantization directly in the data space, rather than only in the latent space? If not, could you comment on the potential concern or provide insights?

---

> ### Author Response · Authors · 2025-11-14
> **Response to the comments of Reviewer ym9y part I**
>
> * __W1. Is there a risk that these parameters become extremely small when needed again__: Thanks for pointing that out. We check the code and find that all the parameters are clipped by $(1e-3,1e3)$ after update. We will include this clipping in paper for clarity.
>
> * __W2. Further explanation is needed for why Eq. (8) encourages greater codebook usage and entropy, and why Eq. (8) takes its specific form__:
> * That is a very good question, and we are still working on a rigorous proof. For now, we can only provide an intuitive explanation.
>     First, we are quantizing $\mu$. For a fixed $D_{KL}=C$, $\mu$ is a bounded random variable with the bound:
>     $$
>     -\sqrt{2C} \le \mu \le \sqrt{2C}.
>     $$
>     Otherwise, we cannot find a proper $\sigma>0$ to achieve $D_{KL}=0.5(\mu^2+\sigma^2-\log \sigma^2 - 1.0)=C$, as $\sigma^2-\log\sigma^2\ge 1$. On the other hand, the pure Gaussian noise-based codebook may have $c_j$ that is significantly larger than $\sqrt{2C}$ or smaller than $-\sqrt{2C}$. These samples are not chosen during quantization and become "dead" vectors in the codebook. The intuition behind Eq. 8 is to encourage the usage of such samples that are too far from $0$, and the simplest regularization is to add a $-||c_j||$ term to promote the selection of these tokens.
>
> * __W3. The connection between IsoKL, MIRACLE, and the proposed method at line 246 is not sufficiently clear__: We will extend the discussion to: "...Additionally, TDC is closely related to the MIRACLE heuristic and the IsoKL parametrization of Gaussian VAE, which also attempt to train a VAE with controlled variation in $D_{KL}$. More specifically, MIRACLE also proposes adjusting $\lambda$ during VAE training. However, MIRACLE only maintains a single $\lambda$, making it effective for controlling the mean of $D_{KL}$ but less effective for constraining its minimum and maximum values. On the other hand, IsoKL imposes strict control on $D_{KL}$ by directly solving for $\sigma$ given $\mu$ using the Lambert W function. However, IsoKL suffers from numerical issues and leads to suboptimal performance."
>
> * __W4. the quantization error is measured in latent space and distortion caused by quantization in data space is not discussed__: We first formalize the condition in data space.
>     * __Corollary 3__ Following the setting in __Theorem 1__, and assuming the decoder $g(.)$ satisify $|g(x_1)-g(x_2)| \le c_3 |x_1 - x_2|$, we have:
>     $$
>     Pr(|g(\hat{z}_i)-g(\mu_i)|\ge c_3\sigma_i) \le \exp(-e^{t}\sqrt{\frac{2}{\pi}}e^{-c_1-0.5}).
>     $$
>     * proof: as $|g(\hat{z}_i)-g(\mu_i)| \le c_3 |\hat{z}_i - \mu_i|$, we have $Pr(|g(\hat{z}_i)-g(\mu_i)|\ge c_3\sigma_i) \le Pr(c_3 |\hat{z_i}-\mu_i|\ge c_3 \sigma_i) = Pr(|\hat{z_i}-\mu_i|\ge \sigma_i)$.
>
>     We can see that theoretically, the quantization error can be magnified by the Lipschitz constant $c_3$. However, we note that this is not a significant issue in practice. As shown in the table below, the actual loss of quality caused by GQ remains reasonable.
>
>     |            | PSNR | LPIPS | SSIM | rFID |
>     |------------|------|-------|------|------|
>     | g($\mu$) | 32.92 | 0.020 | 0.913 | 0.46 |
>     | g(z), $z\sim q(Z\|X)$ | 32.61 | 0.021 | 0.911 | 0.46 |
>     | g($\hat{z}$), $\hat{z}$ from GQ | 32.11 | 0.023 | 0.906 | 0.414 |
>
> * __W5. The definition of distortion in Eq. (1) is unclear__: Eq. 1 is the abstract rate distortion target of VQ-VAE, we will extend Eq. 1 into:
>     $$\log K + E[\Delta(X,g(\hat{z}))] + \mathcal{L}\_{reg},$$
>     and explain that $\mathcal{L}_{reg}$ may be composed of the commitment loss and codebook loss from [Neural discrete representation learning], or the Gumbel softmax loss from [Continuous Relaxation Training of Discrete Latent Variable Image Models].
>
> * __W6. ...Theorem 1, both i and j are used. Isn't this a typo?__: Yes indeed, thanks for pointing that out and we will fix this typo.

---

> ### Author Response · Authors · 2025-11-14
> **Response to the comments of Reviewer ym9y part II**
>
> * __Q1. W1__: See W1. Besides, we performed an additional ablation study, covering tolerances of 0.1, 0.5, and 1.0, and update parameters of 0.999/1.001, 0.99/1.01, and 0.9/1.1. The results, shown below, lead to the overall conclusion that our GQ method is quite robust and not highly sensitive to update scale. On the other hand, reducing the tolerance to $0.1$ does not effect the result much, while increasing the tolerance to $1.0$ has negative effect.
>
>     | method | tolerance | update scale | bpp (num of tokens) | PSNR | LPIPS | SSIM | rFID |
>     |----|-----------|-------------|---------------------|------|-------|------|------|
>     | GQ | 0.5       | 0.99/1.01   | 0.25 ($2^{16}\times 1024$) | 27.61 | 0.059 | 0.807 | 0.529 |
>     | GQ | 0.1       | 0.99/1.01   | 0.25 ($2^{16}\times 1024$) | 27.56 | 0.058 | 0.812 | 0.551 |
>     | GQ | 1.0       | 0.99/1.01   | 0.25 ($2^{16}\times 1024$) | 27.61 | 0.063 | 0.811 | 0.701 |
>     | GQ | 0.5       | 0.9/1.1     | 0.25 ($2^{16}\times 1024$) | 27.63 | 0.060 | 0.809 | 0.534 |
>     | GQ | 0.5       | 0.999/1.001 | 0.25 ($2^{16}\times 1024$) | 27.48 | 0.058 | 0.804 | 0.510 |
>
> * __Q2. give further explanations regarding W2 W3__: See W2, W3.
>
> * __Q3. Have you considered evaluating the distortion caused by quantization directly in the data space, rather than only in the latent space? If not, could you comment on the potential concern or provide insights?__: See W4.

---

> > ### Comment · Reviewer_ym9y · 2025-11-26
> >
> > Thank you for your response. Although my concerns have been mostly resolved, I have a few additional comments.
> >
> > **Regarding the answer to W4:**
> > Thank you for providing the corollary using the Lipschitz constant of the decoder.
> > I would also use the Lipschitz constant if I had to answer immediately.
> > I had expected that the explicit use of Lipschitzness might be avoided by considering the distortion loss in the objective of Gaussian VAE, but it seems difficult.
> > However, from an empirical perspective, Table 16 sufficiently shows the effect of quantization in pixel space and this is an acceptable analysis of the error in pixel space. Therefore, my concern on this point has been largely addressed.
> >
> > **Regarding the answer to W2:**
> > I think $D_{KL}(Z_i \mid X)$ and each $\mu$ depend on the input variable $X$ and thus $\sqrt{2D_{KL}}$ and $\mu$ can be sufficiently large depending on $x$ and trained Gaussian VAE.
> > In addition, we can expect that $D_{KL(2)}(q(Z_i \mid X) || N(0,1))$ is close to $\log_2 K$ when TDC is applied to training of Gaussian VAE, preventing $\sqrt{2D_{KL}}$ from becoming too small .
> > For these reasons, I think the current revised explanation still needs further modification and clarification.
> >
> > Additionally, I would recommend clarifying that $\alpha$ and $\beta$, which were introduced in the revised part, are hyperparameters of the proposed algorithm, for readability.
> >
> > At any rate, my concerns have been almost addressed. I maintain my positive score.

---

> > > ### Author Response · Authors · 2025-12-01
> > >
> > > * __I think $D\_{KL}$ can be sufficiently large preventing__ $\sqrt{2D_{KL}}$ __from too small__: Thanks for the further clarification. $D_{KL}$ is sufficiently large is only true for group=1 case. However for group>1 case, it can be true that for some dimension in the group, the $D_{KL}$ allocated to this dimension is small. This is why we only use Eq. (8) for group>1 case.
> > >
> > > * And for sure, we will clarify $\alpha,\beta$ in revised paper.

---

### Official Review · Reviewer_uBPL · 2025-10-31

**Soundness:** 3
**Presentation:** 2
**Contribution:** 3
**Rating:** 6
**Confidence:** 3

**Summary:**

This paper proposes Gaussian Quant (GQ), a method that converts a Gaussian VAE into a VQ-VAE without additional training. GQ achieves training-free conversion by using noise sampled from a Gaussian distribution as the codebook and selecting the codebook entry closest to the posterior mean of the Gaussian VAE. Theoretically, the authors prove that when the logarithm of the codebook size exceeds the bits-back coding rate, the quantization error becomes small. Furthermore, the paper proposes Target Divergence Constraint (TDC) as a technique to train Gaussian VAEs for effective GQ conversion, demonstrating that this is a simple yet powerful quantization method by matching the KL divergence of each dimension to the bits-back coding rate.

**Strengths:**

- While the idea of converting a Gaussian VAE trained in continuous space into a VQ-VAE is simple, the paper clearly motivates the story by first showing that direct conversion from a vanilla Gaussian VAE does not perform well, and then demonstrating that TDC enables training of Gaussian VAEs suitable for quantization.

- The paper establishes a theoretical relationship between codebook size and bits-back coding rate. By proving theorems showing that quantization error decreases doubly exponentially when $\log K$ matches or exceeds the bits-back coding rate (Eq. 4), and increases exponentially when it falls below (Eq. 5), the work provides principled guidelines for "how to choose $K$".

- The paper compares against existing discrete tokenizers (VQGAN, FSQ, LFQ, BSQ) on ImageNet at identical bpp settings, demonstrating superior performance across all metrics: PSNR, LPIPS, SSIM, and rFID. Moreover, the effectiveness is confirmed on both UNet and ViT architectures, demonstrating generality that does not depend on specific backbones.

**Weaknesses:**

- The core claim of the method is that "a pre-trained Gaussian VAE can be converted to VQ without additional training," but in practice, applying GQ to a Gaussian VAE trained without TDC constraint results in significant performance degradation (PSNR: 26.43 dB vs 32.11 dB in Table 6). In other words, "GQ is training-free" only applies to the conversion step itself, and there is a prerequisite of "training a Gaussian VAE with TDC" in the preceding stage. While the paper mentions conversion from a "pre-trained Gaussian VAE" and indicates that pre-training of the Gaussian VAE is necessary, the critical importance of the TDC constraint should have been emphasized more strongly.

- From the perspective of experimental scope diversity, there are the following limitations:
    - Image generation task evaluation (gFID, IS) is limited to ImageNet, with only reconstruction performance evaluated on COCO. Validation of generation performance on more diverse datasets is desired.
    - The evaluation is mainly conducted in the 0.22-1.00 bpp range, as acknowledged in the paper: "We limit our evaluation... to 0.22–1.00 bpp, which extends the BSQ's original range of 0.25–0.50 bpp." Performance in lower bpp regions ($\leq 0.2$) and at higher resolutions remains as future work.

- TDC depends on multiple hyperparameters ($\lambda\_{\min}$, $\lambda\_{\text{mean}}$, $\lambda\_{\max}$) and dynamic update heuristics (update rate of 1.01, threshold of ±0.5 bits as shown in Eq. 7), but there is insufficient sensitivity analysis or theoretical justification for these parameter choices. In particular, robustness across different architectures and datasets has not been sufficiently validated, which remains an important challenge for practical deployment.

**Questions:**

- What is the rationale for choosing the TDC hyperparameters (especially the ±0.5 bits threshold and the 1.01 update rate)? Is there any sensitivity analysis or performance comparison with different settings for these values?

---

> ### Author Response · Authors · 2025-11-14
> **Response to the comments of Reviewer uBPL part I**
>
> Thank you for your detailed review. We are pleased to provide answers to your questions:
>
> * __W1. ...GQ is training-free" only applies to the conversion step itself, and there is a prerequisite of "training a Gaussian VAE with TDC...__: We amend the statement to: "a pre-trained Gaussian VAE with a specific constraint can be converted to a VQ model without additional training." The point of this paper is that a Gaussian VAE is easier to train than a VQ-VAE; therefore, it is valuable to convert a Gaussian VAE into a VQ-VAE, even when the Gaussian VAE requires training with a specific constraint. In Table 2, we clearly show that GQ outperforms other VQ-VAEs with significantly less training required.
>
> * __W2. ...Validation of generation performance on more diverse datasets is desired...__: Our current focus is on vector quantization rather than image generation, as a tokenizer's performance is not strongly correlated with generation quality [Learnings from scaling visual tokenizers for reconstruction and generation]. Generation performance is primarily affected by how well the data aligns with certain contrastive learning features, such as DINO. Moreover, reconstruction has been shown to be potentially detrimental to generation [Image understanding makes for a good tokenizer for image generation; Reconstruction vs. Generation: Taming Optimization Dilemma in Latent Diffusion Models; GigaTok: Scaling Visual Tokenizers to 3 Billion Parameters for Autoregressive Image Generation]. Our GQ method is orthogonal to approaches aimed at improving generation performance. We plan to investigate GQ with feature alignment to optimize for generation in future work.
>
> * __W3. The evaluation is mainly conducted in the 0.22-1.00 bpp range, as acknowledged in the paper...__: We provide additional results in the low bits-per-pixel (bpp) range ($\le 0.1$), which demonstrate that GQ remains competitive in this regime. We plan to investigate low bpp performance and higher-resolution inputs in future work.
>
>     |                           | bpp (num of tokens) | PSNR  | LPIPS | SSIM  | rFID  |
>     |---------------------------|---------------------|-------|-------|-------|------|
>     | OpenMagViT-V2 (LFQ) | 0.07 ($2^{18}\times 256$)    | 21.63 | 0.111 | 0.640 | 1.17 |
>     | ReVQ-256T | 0.07 ($2^{18}\times 256$)    | 21.96 | 0.121 | 0.640 | 2.05 |
>     | GQ (Ours)                 | 0.07 ($2^{18}\times 256$)   | 22.30 | 0.116 | 0.642 | 1.04 |

---

> ### Author Response · Authors · 2025-11-14
> **Response to the comments of Reviewer uBPL part II**
>
> * __W4. ...but there is insufficient sensitivity analysis or theoretical justification for these parameter choices. In particular, robustness across different architectures and datasets has not been sufficiently validated...__: Thank you for the suggestions. The general intuition for choosing a threshold of 0.5 is that any value within a range of $D_{KL} \pm 0.5$ will be quantized into the same number. The update rate is based on the limit $(1+1/n)^n \approx e$. We selected $n=100$ so that every 100 steps, $\lambda$ can change by at most a factor of $e$. A value of $n$ that is too large might slow down training, while one that is too small might lead to stability issues.
>
>     To further investigate this problem, we performed an additional ablation study, covering tolerances of 0.1, 0.5, and 1.0, and update parameters of 0.999/1.001, 0.99/1.01, and 0.9/1.1. The results, shown below, lead to the overall conclusion that our GQ method is quite robust and not highly sensitive to update scale. On the other hand, reducing the tolerance to $0.1$ does not effect the result much, while increasing the tolerance to $1.0$ has negative effect.
>
>     | method | tolerance | update scale | bpp (num of tokens) | PSNR | LPIPS | SSIM | rFID |
>     |----|-----------|-------------|---------------------|------|-------|------|------|
>     | GQ | 0.5       | 0.99/1.01   | 0.25 ($2^{16}\times 1024$) | 27.61 | 0.059 | 0.807 | 0.529 |
>     | GQ | 0.1       | 0.99/1.01   | 0.25 ($2^{16}\times 1024$) | 27.56 | 0.058 | 0.812 | 0.551 |
>     | GQ | 1.0       | 0.99/1.01   | 0.25 ($2^{16}\times 1024$) | 27.61 | 0.063 | 0.811 | 0.701 |
>     | GQ | 0.5       | 0.9/1.1     | 0.25 ($2^{16}\times 1024$) | 27.63 | 0.060 | 0.809 | 0.534 |
>     | GQ | 0.5       | 0.999/1.001 | 0.25 ($2^{16}\times 1024$) | 27.48 | 0.058 | 0.804 | 0.510 |
>
>     In terms of model architectures, we have validated GQ and its parameter choices on both the SD3 VAE's UNet and the BSQ ViT architecture. Regarding datasets, we have additionally trained GQ on the LAION 2B dataset with an aesthetic score ≥ 5, while still using the ImageNet validation set for testing. The results show that using the LAION dataset does not affect GQ's performance significantly, indicating the robustness of our approach.
>
>     | method | Training Datset      | bpp (num of tokens) | PSNR | LPIPS | SSIM | rFID |
>     |--------|----------------------|---------------------|------|-------|------|------|
>     | GQ     | ImageNet train split | 0.25 ($2^{16}\times 1024$) | 27.61 | 0.059 | 0.807 | 0.529 |
>     | GQ     | LAION 2B (aes >= 5)  | 0.25 ($2^{16}\times 1024$) | 27.77 | 0.060 | 0.812 | 0.626 |
>
> * __Q1. What is the rationale for choosing the TDC hyperparameters__ See W4.

---

> > ### Comment · Reviewer_uBPL · 2025-11-27
> >
> > Thank you for your detailed additional experiments and thoughtful responses. While several of my concerns have been addressed, I would like to comment on points that remain open for discussion.
> >
> > ### **Regarding W2 (Validation of Generation Performance)**
> >
> > I **still cannot agree** with your position that "the focus is on vector quantization, and generation performance is a future challenge."
> >
> > - In your response to W1, you state that "converting a Gaussian VAE to a VQ-VAE is valuable"
> > - However, the primary application of VQ-VAE is **precisely generation tasks** (autoregressive generation, MIM)
> >
> > The literature you cite demonstrates that "reconstruction performance and generation performance are not necessarily correlated," but this actually **emphasizes the necessity of demonstrating that GQ does not impair generation performance**. This is because superior reconstruction does not guarantee superior generation—it may in fact be inferior.
> >
> > Furthermore, regarding your claim that "the GQ method is orthogonal to approaches aimed at improving generation performance," it is precisely because of this orthogonality that **validation of not impairing the final generation quality, which is most important to end users, is essential**. Given that the primary application of VQ-VAE is generation tasks, the insufficient validation on this point undermines the completeness of the paper.
> >
> > Nevertheless, I recognize the value and effectiveness of the paper itself, as well as its potential for future development, and therefore maintain my score.

---

> ### Author Response · Authors · 2025-11-28
> **Response to the comments of Reviewer uBPL part III**
>
> * __Validation of generation performance on different datasets__: Thanks for the further clarification. Now we understand that generation is important as it is the most important application. For different datasets, we test the performance of BSQ, GQ and TokenBridge on FFHQ dataset with the same setup as ImageNet. The result is consistent with ImageNet dataset and shown below. Our approach is quite competitive compared to two most recent variant of VQ-VAE (BSQ and TokenBridge).
>
>     |             | bpp (num of token)        | gFID | IS |
>     |-------------|---------------------------|------|----|
>     | TokenBridge | $0.25 (2^{16}\times 256)$ | 5.48 | - |
>     | BSQ | $0.25 (2^{16}\times 256)$ | 7.15 | - |
>     | GQ | $0.25 (2^{16}\times 256)$ | 5.09 | - |

---

### Official Review · Reviewer_TXyD · 2025-10-31

**Soundness:** 3
**Presentation:** 2
**Contribution:** 3
**Rating:** 6
**Confidence:** 3

**Summary:**

This paper proposes Gaussian Quant (GQ), a training-free procedure that converts a pretrained Gaussian VAE into a VQ-VAE. GQ generates random Gaussian noise as a codebook and quantizes each dimension of the posterior mean to the closest codeword. The authors prove that when the codebook capacity exceeds the VAE's bits-back rate, the induced quantization error is small. They further introduce a Target Divergence Constraint (TDC) to train VAEs that convert more effectively. Experimental results demonstrate that GQ outperforms previous VQ-VAE variants and Gaussian VAE discretization methods.

**Strengths:**

- The idea is interesting and theoretically grounded.

- Practical method TDC is a lightweight tweak that improves the downstream discretization quality.

- Empirical results and ablations are reported across architectures.

**Weaknesses:**

- The paper discusses “grouping to multiple dimensions” for very low‑bitrate regimes, but it’s unclear whether grouping is evaluated as a separate ablation in the main experiments. Clarifying this would help.

- As GQ quantizes each dimension of the posterior mean to the closest codeword, this results in much longer discrete token sequences compared to VQ-VAE variants that quantize sub-vectors, making the generation step more challenging and computationally expensive.

**Questions:**

I would like to clarify:
- Do the pretrained baselines in Table 2 use different architectures? And in Table 1, do all baselines and the proposed GQ method use the same architecture?
- Does GQ generate a single set of K codewords from a unit Gaussian N(0,1) and use it for all posterior-mean dimensions? It would be interesting to see some geometric analysis of the quantized latents, i.e., t-SNE visualizations of some classes.

---

> ### Author Response · Authors · 2025-11-13
> **Response to the comments of Reviewer TXyD**
>
> Thank you for your detailed review. We are pleased to provide answers to your questions:
>
> * __W1. ...it’s unclear whether grouping is evaluated as a separate ablation in the main experiments...__: We performed an ablation study on grouping in Appendix D and Table 12. The overall conclusion is that training-aware grouping (TR) performs best, and selecting a large number of groups improves reconstruction performance. In the main experiments, we set the latent dimension of Gaussian VAE to $\log K$ and set the group size of $\log K$. For example, for a VQ-VAE with a codebook size of $2^{16}$, we train a Gaussian VAE with a latent dimension of 16 and group all 16 dimensions into a single token with a range of $0$ to $2^{16}-1$.
>
> * __W2. ...As GQ quantizes each dimension of the posterior mean to the closest codeword, this results in much longer discrete token sequences compared to VQ-VAE variants that quantize sub-vectors, making the generation step more challenging...__: In image generation, for an image latent of size (C, H, W), even when tokenization is applied to each dimension of C, it is common practice to group all channel dimensions together and run the auto-regressive model on the (H, W) dimensions. This is true for most previous works, such as the open-sourced LFQ (Open-MAGVIT2: An Open-Source Project Toward Democratizing Auto-regressive Visual Generation) and BSQ (Image and Video Tokenization with Binary Spherical Quantization). Although the subcodebooks of LFQ and BSQ are binary, they still group all tokens across the channel into one large token with a range of $2^{14}$ to $2^{18}$, much like the traditional VQ-VAE. This is likely because using smaller subtokens and running an auto-regressive model over (C, H, W) is challenging, as discussed in Section 3.2 of "Bridging Continuous and Discrete Tokens for Autoregressive Visual Generation."
>
> * __Q1. Do the pretrained baselines in Table 2 use different architectures? And in Table 1, do all baselines and the proposed GQ method use the same architecture?__: In Table 1, all methods use the same model architecture; the methods on the left all use the standard SD3.5 VAE's UNet, while those on the right all use the ViT from BSQ. In Table 2, the other methods may use their own specific model architectures, whereas our GQ method adheres to the SD3.5 VAE's UNet. To account for the difference in model size, we have extended Table 2 to include parameter counts. The results show that GQ's parameter count is either smaller than or comparable to the other methods, making it clear that GQ's advantage does not stem from a larger model.
>
>
>     |              | bpp (num of tokens)        | PSNR | LPIPS | SSIM | rFID | Training Epochs | Params (M) |
>     |--------------|----------------------------|------|-------|------|------|-----------------|------------|
>     | VQGAN-Taming | 0.22 ($2^{14}\times 1024$) | 23.38 | - | - | 1.190 | (OpenImages) | 67         |
>     | VQGAN-SD     | 0.22 ($2^{14}\times 1024$) | - | - | - | 1.140 | (OpenImages) | 83         |
>     | Llama-gen-32 | 0.22 ($2^{14}\times 1024$) | 24.44 | 0.064 | 0.768 | 0.590 | 40 | 70         |
>     | FlowMo-Hi    | 0.22 ($2^{14}\times 1024$) | 24.93 | 0.073 | 0.785 | 0.560 | 300 | 945        |
>     | GQ           | 0.22 ($2^{14}\times 1024$) | 25.31 | 0.064 | 0.762 | 0.491 | 40 | 82         |
>     | BSQ          | 0.28 ($2^{18}\times 1024$) | 27.78 | 0.063 | 0.817 | 0.990 | 200 | 175        |
>     | GQ           | 0.28 ($2^{18}\times 1024$) | 27.86 | 0.054 | 0.804 | 0.424 | 40 | 87         |
>
> * __Q2. Does GQ generate a single set of K codewords from a unit Gaussian N(0,1) and use it for all posterior-mean dimensions?__: Yes, the codebook is the same across all dimensions. This is a requirement of VQ-VAE.
>
> * __Q3. It would be interesting to see some geometric analysis of the quantized latents, i.e., t-SNE visualizations of some classes.__: Please find the t-SNE visualizations of 5 ImageNet classes at the provided anonymous link https://ibb.co/d00KwLzM.

---

> > ### Comment · Reviewer_TXyD · 2025-11-26
> >
> > Thank you for your response and clarification. My concerns have been almost addressed, and I maintain my positive score.
> >
> > I have one additional point regarding the quantized latent visualization. This does not affect my positive evaluation. I would expect some comparison or discussion of the latent space differences between GQ and the baselines. If there are interesting insights, this would benefit both the paper and its readers. I'm not trying to create additional work, but a visualization without discussion provides limited value. The decision to include it depends on the authors, if it doesn't support the claims or provide informative insights, it may be better to omit it.

---

> > > ### Author Response · Authors · 2025-12-01
> > >
> > > * __Comparison or discussion of the latent space differences between GQ and the baselines__ : Thanks for the additional clarification. In https://ibb.co/SzkM9YJ we provide the t-SNE of latent space of Gaussian VAE and GaussianQuant. It is shown that quantization does not change latent space significantly, which supports our claim that GaussianQuant does not deviate much from Gaussian VAE.

---

### Author Response · Authors · 2025-11-16
**Summary of Revisions**

Thank you for your detailed review. We have uploaded the revised paper, with all the revisions marked in blue. Below is a summary of revisions:
* We emphasis that our Gaussian VAE requires specific constraint, as suggested by uBPL.
* We clarify the definition of VQ-VAE loss in Eq.1, and GQ codebook is dimension independent as suggested by ym9y and MSjv.
* We include ablation study on TDC parameters in Table 12, as suggesed by uBPL and ym9y.
* We clarify grouping related questions, as suggested by ym9y and MSjv.
* We explain IsoKL, MIRACLE and TokenBridge in main text, as suggested by ym9y and MSjv.
* We include model parameter in Table 2, TokenBridge generation in Table 4, and low bpp result in Table 11 as suggested by uBPL, TXyD and MSjv.
* We include generation result for different bitrates and datasets as suggested by uBPL and MSjv.
* We include results and discussion on the quantization loss in pixel space in Table 16, as suggested by ym9y.
* We include result of codebook with different seed in Table 17, result of simply increase codebook size in Table 18, and t-SNE visualization in Figure 4 as suggested by MSjv and TXyD.
* We fix the typos as suggested by ym9y and MSjv.

---

### Author Response · Authors · 2025-12-02
**Summary of Reviewers' Concern and Rebuttal Action**

We sincerely thank the Area Chair for taking on the extra responsibility and workload resulting from the recent review reset. To facilitate their work under these circumstances, we have prepared this brief overview of the final reviewer positions and the resolutions reached during the discussion period.

The discussion confirmed that three of the four reviewers explicitly stated positive final rate in their last response. The only reviewer with a negative rating did not give a final rating before the interruptation of discussion in 28 Nov, but we have thoroughly addressed all the concerns. A brief overview is provided in the table below:

| Reviewer | Main Concern                                                | Rebuttal Action / Fix                                                                                   | Last Response                            |
|----------|-------------------------------------------------------------|---------------------------------------------------------------------------------------------------------|------------------------------------------|
| TXyD     | Ablation of grouping strategy | Clarified that an ablation study is already provided in Table 12. | Explicitly stated positive final rating. |
| - | Impact of model size                                        | Provided additional results on model parameter size.	 | - |
| - | Visualization of latent space.                              | Provided additional visualizations of the latent space.	 | - |
| uBPL | The importance of TDC constraint should be emphasized more. | Amended the text to emphasize the TDC constraint more strongly. | Explicitly stated positive final rating. |
| - | Generation performance on more diverse datasets.            | Provided generation results on additional datasets. | - |
| - | The evaluation is limited to high bitrate.                  | Provided results for a low bitrate scenario.  | - |
| - | The TDC parameter choice.                                   | Provided an ablation study on the TDC parameter choice.	 | - |
| ym9y     | The TDC update policy and parameter choice.          | Provided clarification and an ablation study on the TDC parameter choice. | Explicitly stated positive final rating. |
| - | Why Eq. 8 is needed.                                        | Provided further clarification in the text. | - |
| - | More discussion on IsoKL, MIRACLE and TDC.                  | Expanded the discussion as requested. | - |
| - | Theoretical result in pixel space.                          | Provided the theoretical result and empirical validation in pixel space. | - |
| MSjv     | Advantage of VQVAE over Gaussian VAE. | Compared the generation performance of autoregressive using VQVAE and diffusion models using Gaussian VAE. | Not replied but concerns addressed.      |
| - | Comparison with TokenBridge and LFQ.                        | Provided generation performance for TokenBridge and reconstruction performance for OpenMagViT-V2 (LFQ).	 | - |
| - | Generation performance on different datasets and bitrates.  | Provided generation performance on different datasets and at various bitrates. | - |
| - | Prior-posterior mismatch.                                   | Provided empirical validation showing the prior-posterior mismatch is not severe. | - |

We hope this summary aids in the final decision-making process.

---

### Meta-Review · Area_Chair_c8Da · 2025-12-17

**Summary:**

The paper introduces Gaussian Quant (GQ), a method that converts a Gaussian VAE trained with a target divergence constraint (TDC) into a VQ-VAE without requiring additional training. While reviewers generally agree that the idea is interesting and the reconstruction results are compelling, several concerns have been raised:
- The primary motivation for this work is to convert a Gaussian VAE into a VQ-VAE, which is typically applied to autoregressive generation tasks. However, the paper mainly focuses on image reconstruction tasks, which limits the scope of the contribution.
- The paper’s presentation lacks clarity and precision, which led to several rounds of discussions between some reviewers and authors to address ambiguities and missing details.

Overall, while the paper demonstrates some technical merit, its experimental scope is somewhat limited, and the presentation could benefit from further refinement. This is a borderline paper, and I encourage the authors to address these issues in a resubmission.

**Reviewer Concerns:**

**Addressed concerns:**
- Several reviewers requested more emphasis on the TDC constraint. In response, the authors revised the manuscript to highlight the TDC's effect more clearly and included additional experimental results.
- Some reviewers raised concerns about posterior-prior mismatch and the authors provided empirical results showing the mismatch is not a significant issue.
- Some reviewers asked for additional latent space visualizations.

**Outstanding concerns:**
- The biggest concern is the positioning/motivation for VQ-VAE over Gaussian VAE. While the authors provided additional generation results on the FFHQ dataset in the rebuttal, the real impact of this work remains unclear. The generation performance of the proposed method still lags significantly behind state-of-the-art models, and reconstruction, which is the focus of this paper, is not the most critical task for VQ-VAE for image-related tasks. The authors should consider providing additional experiments in other data modalities where reconstruction plays a more critical role or providing a more convincing generation comparison.
- While the authors have made some improvements, the paper still lacks clarity and precision. The lack of detail in the original submission led to several rounds of discussion, and some parts of the manuscript could be further refined and illustrated to make the technical details easier to follow.

**Reviewer Scores:**

I think all reviewers would maintain their initial scores. In particular:
- Reviewer uBPL is still concerned about the generation performance but acknowledge the method's strengths.
- Reviewer MSjv raised significant concerns about the motivation and generation performance. While the authors provided additional generation results on a new dataset, the applicability of the proposed method to other tasks where reconstruction plays a more critical role remains unclear.

---

### Decision · Program_Chairs · 2026-01-26

Reject